# Automated Detection and Classification of Returnable Packaging Based on YOLOV4 Algorithm

Matko Glučina [1], Sandi Baressi Šegota [2], Nikola Anđelić [2,*] and Zlatan Car [2]

1 University of Rijeka, Trg Braće Mažuranića 10, 51000 Rijeka, Croatia
2 Faculty of Engineering, University of Rijeka, Vukovarska 58, 51000 Rijeka, Croatia
* Correspondence: nandelic@riteh.hr; Tel.: +385-51-505-662

**Abstract:** This article describes the implementation of the You Only Look Once (YOLO) detection algorithm for the detection of returnable packaging. The method of creating an original dataset and creating an augmented dataset is shown. The model was evaluated using mean Average Precision (mAP), $F1_{score}$, Precision, Recall, Average Intersection over Union (Average IoU) score, and Average Loss. The training was conducted in four cycles, i.e., 6000, 8000, 10,000, and 20,000 max batches with three different activation functions Mish, ReLU, and Linear (used in 6000 and 8000 max batches). The influence train/test dataset ratio was also investigated. The conducted investigation showed that variation of hyperparameters (activation function and max batch sizes) have a significant influence on detection and classification accuracy with the best results obtained in the case of YOLO version 4 (YOLOV4) with the Mish activation function and max batch size of 20,000 that achieved the highest mAP of 99.96% and lowest average error of 0.3643.

**Keywords:** artificial intelligence algorithms; automated system; convolutional neural network; computer vision; YOLOV4

## 1. Introduction

Today, modern society produces a growing amount of waste which has a huge impact on the environment. Some of the disposed waste materials can last in the environment for long periods up to hundreds to even thousands of years (for example plastics) which can cause significant impacts on animals and plants. Waste management is the most important factor for reducing and preventing the possible impact of hazardous waste in nature. The main reason is that every waste management system consists of actions such as collection, transport, treatment, and disposal of waste, which results in a reduction of the waste materials disposed of in nature [1]. Additionally, to reduce disposed waste in nature, along with the waste management system, it is necessary to implement advanced computational methods that, in combination with artificial intelligence (AI), significantly contribute to the detection and classification of harmful waste.

Object detection has had a great impact in the field of computer vision tasks such as image annotation [2], vehicle counting [3], activity recognition [4], face detection [5], etc., in the past decade. It also has had an essential role in scene understanding, which gained a high level of popularity in security, transportation, medical and military applications. In addition to the above examples, it also has multiple applications: autonomous driving [6], people detection in security [7], vehicle detection with AI in transportation [8], medical feature detection in healthcare, and in waste management [9].

Over the last two decades, AI methods have gained momentum in offering alternative computational approaches to solve solid waste management problems. In the case of the detection and classification of recyclable waste, convolutional neural networks (CNN) were mostly utilized.

Mao et al. [10] used the genetic algorithm (GA) for optimization of a fully connected layer, DenseNet121 to improve the classification accuracy of the result ThrashNet [11]

dataset which consists of 2527 images in six categories of waste (glass, paper, cardboard, plastic, metal, and trash). From the obtained results, it was seen that the optimized DenseNet121 achieved the highest accuracy of 99.6%. Adedeji and Wang [12] proposed the intelligent waste material classification, which is based on ResNet-50 and support vector machines (SVM) applied to TrashNet dataset [11]. ResNet-50 was used as a feature extractor while the SVM was used to classify the waste into the specified groups, the obtained model achieved an accuracy score of 87%. Bobulski and Kubanek [13] performed the plastic waste classification using CNN on the following classes: polyethylene terephthalate, high-density polyethylene, polypropylene, and polystyrene. Authors in the research created a unique dataset, they also investigated the different configurations of CNN (15 and 23-layer network), diverse image resolutions (120 × 120, and 227 × 227 pixels), and the influence of different train/test dataset ratios (60:40 up to 90:10). The highest classification accuracy was achieved in case of 23-layer CNN with the image resolution of 227 × 227 pixels of 99.23% at train/test ratio 90:10. Shi et al. [14] proposed the multi-layer hybrid CNN (similar to VggNet with fewer parameters and higher classification accuracy) and tested on TrashNet [11] dataset, for intelligent waste classification to overcome the low classification accuracy and long running times of the majority of CNN-s. In this investigation, the highest classification accuracy achieved was 92.6%. The SVM along with Histogram of Oriented Gradient (HOG) features CNN, and ResNet50 CNN [15] were applied to publicly available waste dataset from Kaggle [16]. The goal was to identify a single object in images and classify it into one of the given categories (cardboard, glass, metal, paper, plastic, and trash). The highest accuracy score was 95.35% and it was achieved using ResNet50 architecture. On the original dataset Altikat et al. [17] applied the four and five-layer deep CNN for classifying the paper, glass, plastic, and organic waste. The investigation showed that the highest classification accuracy of 83% and 76.7% was achieved in the case of organic waste with four and five-layered deep CNN, respectively. In the last few years, the YOLO algorithm has been increasingly used in the detection and classification of non-biodegradable waste. Aishwarya et al. [18] applied the YOLO algorithm to detect non-biodegradable waste from the bins. Classification accuracy was faulty and imprecise with the results of 75% for metal, 65% for plastic, and 60% for glass. Wahyutama and Hwang [19] reported the development of a smart trash bin that separates and collects trash recyclables using a webcam and YOLO algorithm where the classification accuracy of recyclables using YOLO was 91% in an optimal computing environment and 75% when implemented on Raspberry Pi. Lin [20] used the YOLO-Green object detection algorithm for detecting trash in real time. In this paper, the original dataset was developed from real-world trash divided into seven types of solid waste (batteries, clothes, electronic waste, glass, metal, paper, and plastic). The YOLO-Green achieved a classification accuracy of 78.04%. The YOLO-based neural network model with variational autoencoder was employed for automatic detection and classification of recycling waste [21]. The highest classification accuracy was 69.7%.

The superiority of the YOLO algorithm compared to other detection algorithms can be demonstrated by additional research and applications in practice. For example, Kim et al. [22] trained YOLOV4, Single Shot MultiBox Detector (SSD), and Faster-Region Based Convolutional Neural Networks (Faster-RCNN) for the detection and classification of vehicle models. The obtained results showed that the YOLOV4 algorithm has had the best performance (mAP in the amount of 98.19% and FPS in the amount of 82.1). SSD proved to be the fastest (FPS in the amount of 105.14), but the problem comes from the simplified light model (Mobilnet- lightweight model) which frequently failed to detect a vehicle. Faster-RCNN turned out to be the slowest model (FPS in the amount of 36.32), although it was slightly less accurate (mAP valued at 93.40%) than YOLOV4, due to its detection speed it was unable to perform the given task. According to [21], YOLO can be applied as an algorithm for an automatic waste recycling system. The authors trained the modified YOLO (YOLO with variational Autoencoder) and Fast R-CNN model on a custom-made dataset consisting of cans, batteries, and plastic bottles. The obtained results prove that the modified YOLO model with a prediction accuracy rate of 69.70% and a

localization rate of 22.10% again outperforms the competitive Fast R-CNN with an accuracy rate of 71.60% and a localization rate of 8.60%. In the paper, ref. [23] AI-controlled Outdoor Autonomous Trash-Collecting Robot was used for underwater garbage collection. Several object detection algorithms (Mask-RCNN, YOLOV4, and YOLOV4-tiny) were compared and tested for the outdoor waste detection and classification task. The obtained results were evaluated with the best map and the detection time where YOLOV4 and YOLOV4-tiny outperformed the Mask-RCNN in the detection and localization of outdoor trash. YOLOV4 and YOLOV4-Tiny achieved the best mAP of 99.32% and 95.25% with a detection speed of 32.76 and 5.21 ms, while the Mask-RCNN barely achieved the best mAP of 89.10% and had a very slow detection time in the amount of 3973.29 ms. One of the important factors when choosing the optimal algorithm for implementation in an automatic system is the memory of the model itself. Tian et al. [24] trained Faster R-CNN, 4S-YOLOv4, YOLOv4, YOLOv3, SSD, and 4SP-YOLOv4, on a dataset consisting of 6600 images of underwater garbage. The obtained results show that 4SP YOLO requires significantly less memory for obtaining weight factors than Faster R-CNN or SSD. In this particular case, 4SP YOLO requires $10^2$ less memory than SSD and up to $10^6$ times less memory than Faster R-CNN. This fact gives the possibility of implementing the YOLO detection algorithm to a printed circuit board-based microcontrollers such as Raspberry Pi, CaffeLatte, etc.

The goal of this paper is to examine the YOLOV4 algorithm and achieve the optimal performance by varying activation function and max batch size hyperparameters on a custom-made, three-class (plastic, glass, and aluminum) dataset that contains 3788 images (2838 original images and 950 downloaded from Kaggle [25]). Based on the idea of this investigation and the extensive literature overview, the following hypotheses questions arise:

- Is it possible to achieve high classification accuracy with the YOLOV4 algorithm?
- Is it possible to improve classification accuracy by tweaking the specific hyperparameters (max batch size and activation function) of the YOLOV4 algorithm?
- Does the train/test dataset ratio has any influence on YOLOV4 classification accuracy?

The structure of this paper can be divided into the following sections, i.e., Materials and Methods, Results and Discussion, and Conclusions. In Materials and Methods, the dataset description is given as well as the YOLOV4 algorithm and evaluation methods. In Results and Discussion, the results of the trained YOLOV4 algorithm with different hyperparameters are presented as well as discussed. The conclusions section contains a list of conclusions obtained during this investigation that is based on the results and discussion section and provides direct answers to the hypotheses defined in the Introduction section.

## 2. Materials and Methods

In this section, the procedure of dataset development is described, as well as the utilized YOLOV4 algorithm. The first step for training the YOLOV4 algorithm was to create a diverse and equally distributed dataset. With this fact, the description of creating a dataset can be summarized in the following steps:

- gathering of original images of returnable packaging,
- combination of original images to those obtained from Kaggle,
- performing the dataset augmentation to enlarge the number of images to get as many combinations as possible combinations for training, testing, and validation set,
- labeling the bounding boxes for determination of objects inside of selected image in YOLOV4 compatible format, and
- testing the video camera detection in real-time to check the operation of the designed detection algorithm.

The procedure of collecting original images of returnable packaging such as plastic, glass, and aluminum bottles/cans is described in the following subsection.

### 2.1. Dataset Description and Preparation

The data collection strategy over the life of the service/product is essential for the development of a successful project [26] and it cannot be a series of one-off exercises. Every time a user comes in contact with a desired product or service, it would be desirable to collect data from the interaction [27] to gain a constant flow of data to improve products/programs. Upon reaching that level of data collection, each new customer contributes to the size and diversity of the newly created dataset while the advantage of building this strategy and collecting data is that it becomes difficult for competitors to practically replicate the dataset used in these projects [28]. With larger and more up-to-date data the AI becomes better and much more valuable and because of that the first step was to collect, an original unprocessed packaging is shown in Figure 1. As already stated the part of the dataset was photographed in its natural form, without any processing. These images were created in three steps: Initially, the packaging was photographed without modifications to its original forms. After that, thermal and mechanical processing was applied to obtain different images. Finally, multiple packaging objects, in both processed and unprocessed form were grouped and photographed. It should be noted that the packaging was empty, i.e., did not contain any liquid. The packaging was photographed in several positions: upright, lying down, rotated, or shifted concerning the original position, which contributed to possible conditions during the detection.

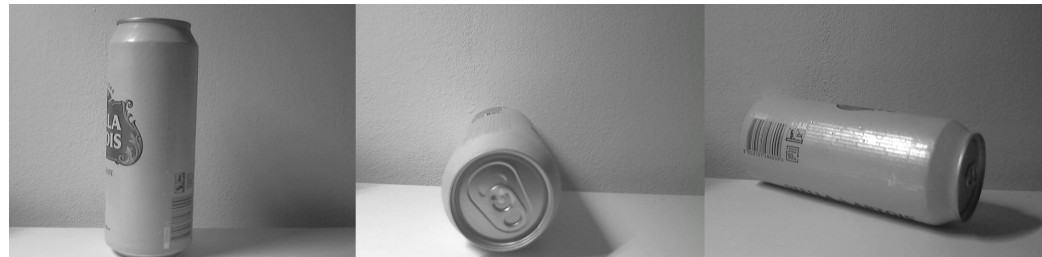

**Figure 1.** An example of the unprocessed dataset.

To expand the dataset, the mechanical and thermal loading on plastic and aluminum packaging was applied to change their shape. The glass packaging could not undergo such modification since it is a hard and brittle material and would shatter under mechanical loading or would crack at higher temperatures. The application of such external loading on packaging greatly increased the dataset size as well as its quality. Additional improvement of dataset quality was conducted by photographing a group of packaging. The examples of packaging deformed by the application of thermal loading, mechanical loading, and a group of all three are shown in Figure 2.

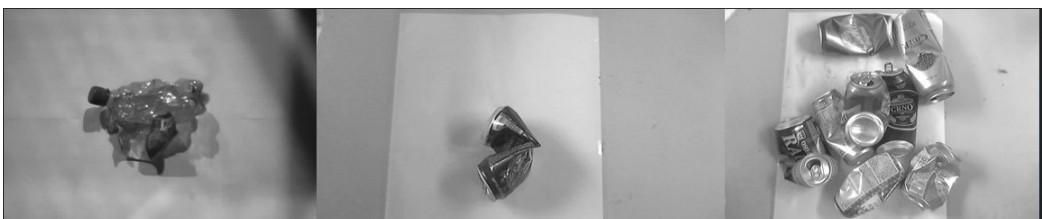

**Figure 2.** Physical transformation actions performed on the dataset.

All the photographs in the dataset are grey-scale because they have small memory size images that have fewer characteristics that can affect the training of YOLOV4 CNN [29,30]. When creating the original dataset, it is important to pay attention that it should be as diverse as possible [31] so that the ML algorithm has plenty different learning examples as possible [32]. The initial dataset consists of 2838 photographs with an additional 950 which are downloaded from Kaggle [25].

The dataset is not sufficient to develop a good AI model [33] and to enlarge it the data augmentation was performed [34,35]. The methods of geometric augmentation [36] that were used are rotation, blurring, flipping, translating, and zooming (scaling) shown in Figure 3.

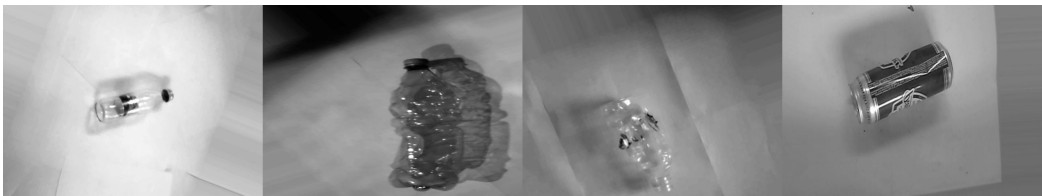

**Figure 3.** Geometric image augmentation for the expansion of the dataset.

The whole dataset collection was performed using OpenCV [37] library in Python programming language. The number of original images, augmented images, and total images in the dataset is presented in Table 1.

**Table 1.** Dataset used in training YOLOV4 algorithm.

| Class | Type of Packaging | Number of Original Images | Number of Augmented Images | Number of Total Images |
|:-----:|:-----------------:|:-------------------------:|:--------------------------:|:----------------------:|
| 1 | Plastic | 1003 | 869 | 1872 |
| 2 | Glass | 1224 | 600 | 1824 |
| 3 | Aluminum | 611 | 694 | 1205 |

In Table 1 the number of original, augmented and total images for each class (plastic, glass, and aluminum) is shown. From the number of original images per class, it can be seen that the dataset was not perfectly balanced [38]. The augmentation was used to compensate for the dataset imbalance and improve its quality [39]. After the original dataset was obtained and augmentation performed the next step was labeling the bounding boxes for the YOLOV4 algorithm using Visual Object Tagging Tool (VoTT) [40]. The marking of masks, i.e., border frames for a single image is shown in Figure 4.

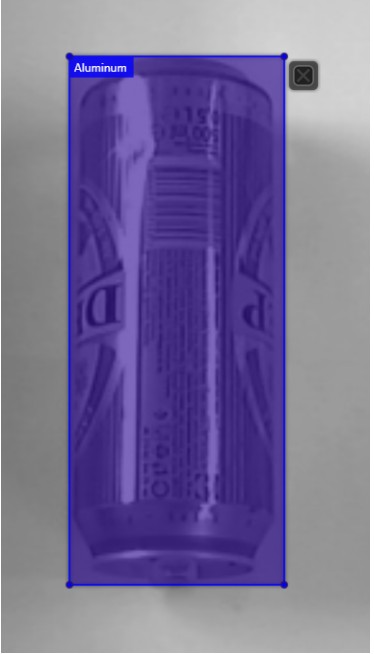

**Figure 4.** Labeling boundary boxes of a single image using VoTT software.

The dataset must be well-prepared [41] for the algorithm to have the best examples available for training, i.e., learning, so the important thing while labeling the dataset was being careful to gather as much vital information as possible and to avoid labeling unnecessary elements [42]. Correct and incorrect labeling of bounding boxes is shown in Figure 5.

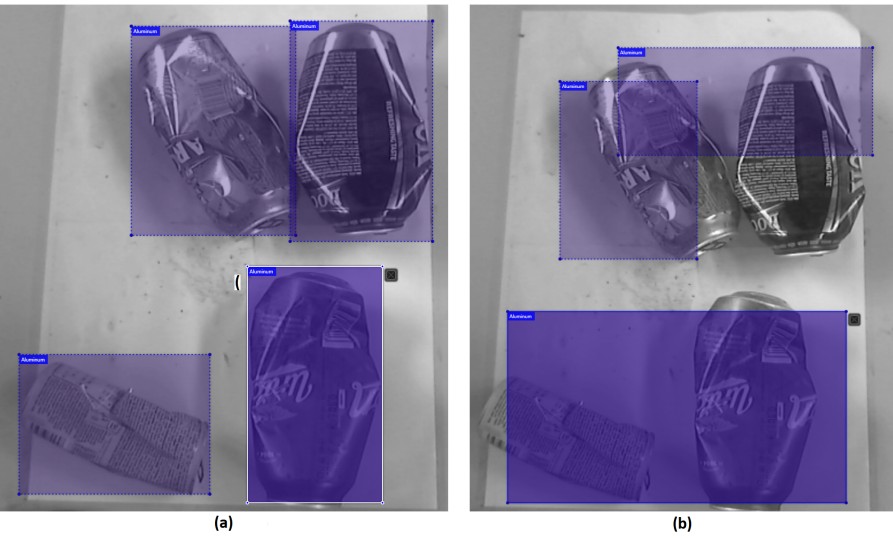

**Figure 5.** Labeling the dataset (**a**) correct way, (**b**) the incorrect way.

In Figure 5, the correct (Figure 5a) and incorrect (Figure 5b) labeling of bounding boxes is shown. Incorrect labeling of bounding boxes (Figure 5b) shows that additional elements such as an empty workspace surface were taken into account during labeling or an unnecessa ry additional element which of course, affects the training of the YOLOV4 algorithm and overall detection. After exporting the VoTT file, the corresponding TXT file is obtained which is shown in Table 2. Each row consists of columns in which each row column denotes a particular property: the first row of the column indicates the image class defined in the VoTT software, and the second and third denote the center of the object along the x and y axes, fourth and fifth denote the width and height of the element [43]. With the successful completion of these actions, the dataset is prepared for training the machine learning algorithm.

**Table 2.** Example of YOLOV4 annotations for Figure 5a TXT file.

| Class | X Center | Y Center | Width | Height |
|-------|----------|----------|--------|--------|
| 1 | 0.2855 | 0.8367 | 0.2593 | 0.2282 |
| 1 | 0.5844 | 0.6422 | 0.2765 | 0.2217 |
| 1 | 0.6008 | 0.3644 | 0.1949 | 0.2619 |
| 1 | 0.2977 | 0.3014 | 0.2495 | 0.2511 |

### 2.2. Description of Methods

YOLOV4 is the most advanced detection algorithm with a real-time object recognition system that can recognize multiple objects in a single frame [44]. It uses a completely different approach from previous detection systems where it applies one neural network to the whole frame, i.e., the image, and then the neural network divides the image into regions and predicts the boundary frames which then calculates the probability for each possible region, i.e., class of objects. According to [44], the basic idea of the YOLOV4 algorithm is the division of input data into a matrix of a certain dimension by which each matrix address is responsible for predicting objects within the center of that cell where then for

every object that algorithm can detect, the probability coefficient is calculated visible from Figure 6.

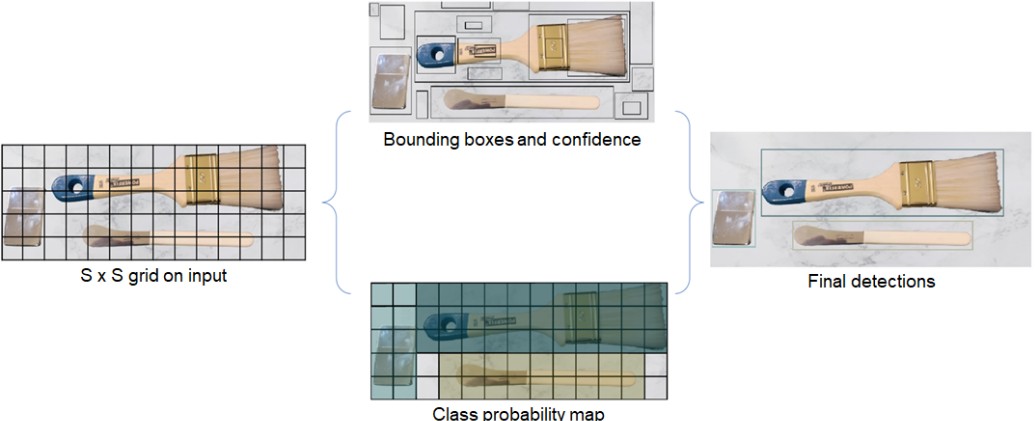

**Figure 6.** An illustration of YOLOV4-based object detection.

As seen in Figure 6, each matrix provides boundary frameworks and calculates a reliability estimate for the corresponding framework. After the reliability estimate is calculated the reliability score is determined which shows how secure the model is and that the frame contains the object [45]. This type of detection algorithm has several advantages: it can recognize several objects in one frame [46], during testing the entire image frame is reviewed so that predictions are based on the global image context and it also provides predictions with a single evaluation of the network as unlikely similar methods such as R-CNN which require thousands of examples of a single image [47,48]. Due to the aforementioned properties, the YOLOV4 algorithm is extremely fast even up to a thousand times faster than R-CNN, and up to a hundred times faster, than Fast R-CNN [49]. The YOLOV4 design allows end-to-end training and real-time speed while maintaining high average accuracy [50] and its flow chart can be seen in Figure 7.

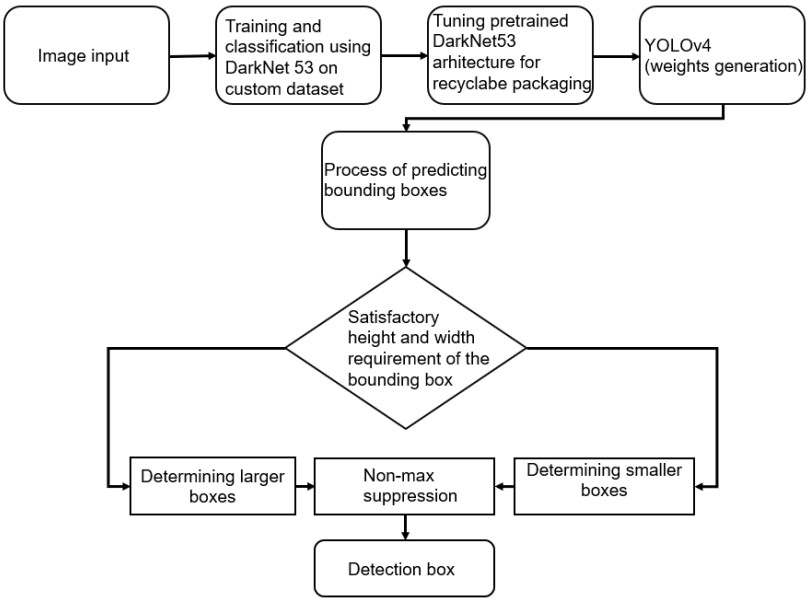

**Figure 7.** Diagram of the YOLOV4 algorithm applied for the detection of recyclable packaging.

The process starts with taking an image from the COCO dataset or custom-labeled dataset where the trained weights are pre-trained so it is necessary to update and adjust,

i.e., fine-tune the weights then the process of predicting bounding boxes begins, which results in large boxes and small boxes where the Non-Max Suppression (NMS) procedure obtains the main box that at the end of detection is the result itself [51,52]. NMS is defined as a local maximum where the local maximum is greater than all of its neighbors except himself. For some n neighborhood, any pixel consists of 1-D case, i.e., n pixel left and right and in the case of the square region, it must be centered around the pixel under consideration. NMS consists of several computer vision algorithms [53], some of them are the measure of the selected point in the whole image, and even the entire scale space, and then the local maximum is selected. Efficiency is a major feature in its applications and the only competitor to NMS is the Maximum filter [54] which calculates the value of each pixel of each neighbor, where the process is much slower than NMS. Given the maximum filter response in each pixel, NMS reduces additional comparative values of each pixel with the largest neighbor. After training the YOLOV4, it is necessary to evaluate the obtained results. There are several suitable metrics, but the most important and generally accepted metrics are mAP and IoU. Using these metrics, the algorithm calculates a reliability coefficient ranging from 0 to 1, i.e., from 0% to 100% confidence that the predicted class from final detection is accurate and true. Depending on the results, it is necessary to either additionally train the algorithm or adjust training parameters to obtain better and more accurate results.

*2.3. YOLOV4 Limitations and Accuracy Measures*

Although YOLOV4 seems to be the best algorithm for detecting application objects, it is essential to pay attention to a few problems and limitations that can occur along the way. The detection algorithm has problems separating small objects in images that appear in groups, although the fourth version of the YOLOV4 has been improved a lot compared to its predecessors [55]. The reason for this is that each neural network is limited to detecting an individual object [22] therefore, the above detection algorithm makes it difficult to detect and localize small objects that are naturally found in group images. IoU is one of the significant metrics needed to measure the accuracy and localization of objects in a frame which can be calculated by finding the area that intersects between the limit frame for a particular prediction and the assumed accuracy limits of that area first taken. The total area covered by these two boundary frames (Union), is then calculated. The intersection divided by the union gives the ratio of the overlap to the total area, which gives a relatively good estimate of how close the boundary frame is to the source prediction, and then after the defined terms, IoU [56–58] can be calculated as Equation (1), where *A* is the area of prediction box and *B* is the area of target box:

$$\text{IoU} = \frac{|A \bigcap B|}{|A \bigcup B|} \tag{1}$$

In Equation (1), the numerator represents the calculation of the intersection/overlap area between the predicted boundary box [59] and the ground-truth bounding box [60] while the Union represents the total union between them. The IoU is in a range from 0 to 100 and the closer the value is to 100 the better it is, as shown in Figure 8.

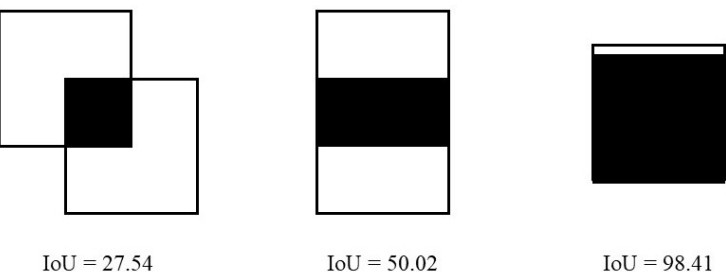

IoU = 27.54          IoU = 50.02          IoU = 98.41

**Figure 8.** Different types of IoU values.

Accuracy is measured with an evaluation metric mAP, which measures in what percentage the algorithm predicted the object from all individual classes correctly and it is expressed in percentages that go from 0% to 100%. The mathematical description of mAP can be represented by Equation (2). Where the $AP_k$ is the average Precision of the classes and the $n$ is the number of classes [61].

$$mAP = \frac{1}{n} \sum_{k=1}^{k=n} AP_k.$$

(2)

The Precision determines how accurate your model is and it can be calculated with the expression [62]:

$$Precision = \frac{TP}{TP + FP}.$$

(3)

The recall is a measure that shows how many times the model finds all the positive cases while training Recall [62] and can be calculated with:

$$Recall = \frac{TP}{TP + FN}.$$

(4)

Finally, the F1$_{score}$ is an evaluation metric that measures models accuracy in combination with Recall and Precision [63] and can be calculated with the expression:

$$F1_{score} = 2 \times \frac{Recall \times Precision}{Recall + Precision},$$

(5)

where:

- TP—True Positive predictions,
- FP—False Positive predictions, and
- FN—False Negative predictions.

YOLOV4 has intrusive spatial constraints in terms of bounding box predictions hence each grid cell can only predict two possible boxes and can only have one unique class. This constraint can limit the number of objects that the model can predict. A model can struggle with small objects that appear in groups such as a flock of birds or a group of ants [64].

Configuration, Setup, and Hyper-Parameters Used for YOLOV4 Project

Before training the YOLOV4 algorithm, it was necessary to define a few configuration files within the YOLOV4 folder. Data folder obj.data contains some vital information where in several lines is described the number of classes that the algorithm must train, the location of the train.txt file that is separated from the total dataset for neural network training, the location of obj.names that containing class names, and test.txt which is used to check or test the neural network. Folders train.txt and test.txt can vary depending on the division of the total dataset into a train set and a test set in the process.py python script. Given that a relatively small set of data was used in this research it was necessary to change a train/test ratio in python script proces.py to get better results. After configuring the above files, the next step was to configure the yolov4-custom.cfg file, which defines max batch size, and set this parameter the total number of training examples present in one series, batch, and subdivision which determines how many images can be processed in parallel, learning rate, etc. All changed hyperparameters (max batch size, activation functions) and train/test dataset ratios are listed in Table 3. By changing, hyperparameters and train/test ratios the results of the evaluation metrics were changed.

As seen from Table 3 there are three activation functions used, i.e., ReLU, Mish, and Linear. Several max batch sizes—6000, 8000, and 10,000—were used for each activation function. However, an additional max batch size of 20,000 was used with the Mish and ReLU activation functions. Due to the relatively small dataset train/test, the ratio has been modified with the most common used train/test ratios that are: 70/30, 75/25, and

80/20 [65]. These ratios were chosen due to the structure, more specifically, the size, of the dataset. Given that it is a small dataset, by choosing a training set ratio greater than 80% or less than 70%, there is a very high probability of overfitting, which is most recommended to be avoided. Another reason the authors used this range of the training dataset (70–80%) is that it is the most commonly used range [66].

**Table 3.** Variation of YOLOV4 algorithm hyperparameters and the size of train/test dataset ratio.

| No. | Max Batches | Activation Function | Train/Test Ratio | No. | Max Batches | Activation Function | Train/Test Ratio |
|---|---|---|---|---|---|---|---|
| 1 | | Linear | | 19 | | ReLU | |
| 2 | | ReLU | | 20 | | Mish | 70/30 |
| 3 | | Mish | 70/30 | 21 | | ReLU | |
| 4 | | Linear | | 22 | 10,000 | Mish | 75/25 |
| 5 | 6000 | ReLU | 75/25 | 23 | | ReLU | |
| 6 | | Mish | | 24 | | Mish | 80/20 |
| 7 | | Linear | | 25 | | ReLU | |
| 8 | | ReLU | | 26 | | Mish | 70/30 |
| 9 | | Mish | 80/20 | 27 | 20,000 | ReLU | |
| 10 | | Linear | | 28 | | Mish | 75/25 |
| 11 | | ReLU | 70/30 | 29 | | ReLU | |
| 12 | | Mish | | 30 | | Mish | 80/20 |
| 13 | | Linear | | | | | |
| 14 | 8000 | ReLU | 75/25 | | | | |
| 15 | | Mish | | | | | |
| 16 | | Linear | | | | | |
| 17 | | ReLU | 80/20 | | | | |
| 18 | | Mish | | | | | |

After the training with specific parameters that have been changed in the configuration file a certain loss [67] occurred as in the expression:

$$
\begin{aligned}
\varepsilon = {} & \lambda_{\text{coord}} \sum_{i=0}^{S^2} \sum_{j=0}^{B} \mathbb{1}_{ij}^{\text{obj}} \left[ (x_i - \hat{x}_i)^2 + (y_i - \hat{y}_i)^2 \right] \\
& + \lambda_{\text{coord}} \sum_{i=0}^{S^2} \sum_{j=0}^{B} \mathbb{1}_{ij}^{\text{obj}} \left[ \left(\sqrt{w_i} - \sqrt{\hat{w}_i}\right)^2 + \left(\sqrt{h_i} - \sqrt{\hat{h}_i}\right)^2 \right] \\
& + \sum_{i=0}^{S^2} \sum_{j=0}^{B} \mathbb{1}_{ij}^{\text{obj}} \left(C_i - \hat{C}_i\right)^2 \\
& + \lambda_{\text{noobj}} \sum_{i=0}^{S^2} \sum_{j=0}^{B} \mathbb{1}_{ij}^{\text{noobj}} \left(C_i - \hat{C}_i\right)^2 \\
& + \sum_{i=0}^{S^2} \mathbb{1}_{i}^{\text{obj}} \sum_{c \in \text{ classes}} \left(p_i(c) - \hat{p}_i(c)\right)^2,
\end{aligned}
\tag{6}
$$

where:

- $x$ and $y$-coordinates of a particular grid in offset in a range from 0 to 1,
- $\lambda_{\text{coord}}$ and $\lambda_{\text{noobj}}$ indicates parameters for increasing the loss value of bounding box coordinates while predicting and decreasing the loss value of confidence predictions bounding boxes that does not appear to contain any objects,
- $\mathbb{1}_{i}^{\text{obj}}$ indicates whether the object appears in cell $i$ and
- $\mathbb{1}_{ij}^{\text{obj}}$ indicates that the $j$-th predictor of the boundary frame in the cell is "responsible" for that prediction.

Notably, the loss function can only penalize the classification error if the represented object is in that specific network cell. Unnecessary problems can possibly occur, so it is

recommended to use the value of $\lambda_{\text{coord}} = 5$ and $\lambda_{\text{noobj}} = 0.5$ [67]. Sum squared error can be also equal to errors in big boxes and small boxes; however, given error metrics should directly reflect on small deviations in big boxes and they matter less than in small boxes. To avoid this problem and confusion bounding boxes for predicting square root error are partially calculated instead of measuring the width and height of boxes directly. YOLOV4 can predict more than one bounding box per fully calculated grid cell. While training, it is necessary to assign a predictor tool that has the task mainly responsible for predicting an object based on its highest IoU with its ground truth. Each possible predictor gets better and it is improving in predicting certain aspects such as size, aspect ratio, or even class of the object shown with expression (6) [67].

## 3. Results and Discussion

In this section, the results of the investigation on the influence of YOLOV4 hyperparameters (max batch size, and activation functions) and train/test ratios on classification accuracy are presented using mAP, Average IoU, F1$_{\text{score}}$, Precision, and Recall metrics. Combinations of hyperparameters and train/test ratios are shown in Table 3 and the key points during the training of the YOLOV4 algorithm are described. In addition, the average loss charts during training and the metric evaluation of the trained model are available in the Appendixes A–D section. The Appendix subsection consists of four categories and three subcategories, while the categories represent each training cycle of 6000, 8000, 10,000, and 20,000 max batch size, and the subcategories represent the Linear ReLU and Mish activation functions. The final table of evaluation metrics can be seen in Tables 4 and 5.

**Table 4.** Overall overview of the results obtained through this research for first two cycles.

| Max Batches | 6000 | | | | | | | | |
|---|---|---|---|---|---|---|---|---|---|
| Activation Function | Linear | | | ReLu | | | Mish | | |
| Train/test ratio | 70/30 | 75/25 | 80/20 | 70/30 | 75/25 | 80/20 | 70/30 | 75/25 | 80/20 |
| mAP | 60.18% | 59.85% | 59.11% | 93.12% | 94.17% | 94.16% | 94.25% | 93.25% | 92.26% |
| F1$_{\text{score}}$ | 0.52 | 0.48 | 0.47 | 0.87 | 0.85 | 0.81 | 0.86 | 0.84 | 0.83 |
| Average IoU | 29.98% | 28.82% | 28.14% | 62.18% | 60.49% | 58.21% | 61.52% | 60.77% | 59.98% |
| Precision | 0.44 | 0.41 | 0.40 | 0.84 | 0.79 | 0.75 | 0.81 | 0.78 | 0.76 |
| Recall | 0.64 | 0.59 | 0.58 | 0.92 | 0.92 | 0.90 | 0.93 | 0.92 | 0.92 |
| Average Loss | 2.6178 | 2.4873 | 2.4285 | 1.1888 | 1.1897 | 1.2689 | 1.1770 | 1.2203 | 1.2406 |
| Max Batches | 8000 | | | | | | | | |
| Activation Function | Linear | | | ReLu | | | Mish | | |
| Train/test ratio | 70/30 | 75/25 | 80/20 | 70/30 | 75/25 | 80/20 | 70/30 | 75/25 | 80/20 |
| mAP | 94.51% | 93.17% | 91.27% | 99.77% | 96.17% | 95.93% | 99.71% | 99.70% | 99.65% |
| F1$_{\text{score}}$ | 0.59 | 0.70 | 0.76 | 0.98 | 0.97 | 0.97 | 0.98 | 0.97 | 0.98 |
| Average IoU | 60.11% | 57.15% | 52.27% | 80.75% | 78.12% | 77.63% | 79.80% | 81.24% | 81.25% |
| Precision | 0.53 | 0.64 | 0.72 | 0.97 | 0.97 | 0.96 | 0.98 | 0.97 | 0.97 |
| Recall | 0.69 | 0.79 | 0.80 | 0.99 | 0.98 | 0.99 | 0.99 | 0.99 | 0.99 |
| Average Loss | 1.8411 | 1.7828 | 2.0389 | 0.9096 | 0.8660 | 0.9215 | 0.9443 | 0.9091 | 0.9145 |

### 3.1. Results

As previously stated, the obtained results are classified into four cycles. Based on the obtained results, it is evident that during the first 1200 iterations the loss curve has a higher value and it is greater than 2000. When reaching the 1200th iteration, there is a rapid reduction of the average loss curve until the 1800th iteration. From that moment on, the loss curve drops linearly until the final iteration. In the last defined iteration of training, the final

value of the average loss is obtained, shown in Tables 4 and 5. The above analysis refers to all trained models of this research. Appendix A represents the average loss charts for 6000, Appendix B for 8000, Appendix C for 10,000, and Appendix D for 20,000 max batches. Appendixes A.1 and B.1 represents Linear activation function. Appendixes A.2, B.2, C.1 and D.1 represents ReLU. The final Mish activation function represents Appendixes A.3, B.3, C.2 and D.2 appendix subsections.

**Table 5.** Overall overview of the results obtained through this research for the third and fourth training cycle.

| Max Batches | 10,000 | | | | | | | | |
|---|---|---|---|---|---|---|---|---|---|
| **Activation Function** | **Linear** | | | **ReLu** | | | **Mish** | | |
| **Train/test ratio** | 70/30 | 75/25 | 80/20 | 70/30 | 75/25 | 80/20 | 70/30 | 75/25 | 80/20 |
| **mAP** | / | / | / | 99.90% | 99.90% | 99.77% | 99.84% | 99.83% | 99.81% |
| **F1$_{score}$** | / | / | / | 0.99 | 0.99 | 0.98 | 0.99 | 0.99 | 0.99 |
| **Average IoU** | / | / | / | 84.14% | 84.59% | 83.41% | 83.44% | 83.59% | 83.33% |
| **Precision** | / | / | / | 0.98 | 0.99 | 0.98 | 0.99 | 0.99 | 0.98 |
| **Recall** | / | / | / | 1.00 | 1.00 | 1.00 | 1.00 | 0.99 | 1.00 |
| **Average Loss** | / | / | / | 0.6320 | 0.6951 | 0.6862 | 0.6945 | 0.6809 | 0.6500 |
| **Max Batches** | 20,000 | | | | | | | | |
| **Activation Function** | **Linear** | | | **ReLu** | | | **Mish** | | |
| **Train/test ratio** | 70/30 | 75/25 | 80/20 | 70/30 | 75/25 | 80/20 | 70/30 | 75/25 | 80/20 |
| **mAP** | / | / | / | 88.33% | 88.60% | 89.08% | 89.92% | 99.96% | 99.94% |
| **F1$_{score}$** | / | / | / | 0.86 | 0.86 | 0.87 | 0.87 | 1.00 | 1.00 |
| **Average IoU** | / | / | / | 84.41% | 86.68% | 83.90% | 82.73% | 91.13% | 91.06% |
| **Precision** | / | / | / | 0.95 | 0.96 | 0.94 | 0.93 | 1.00 | 1.00 |
| **Recall** | / | / | / | 0.75 | 0.79 | 0.80 | 0.81 | 1.00 | 1.00 |
| **Average Loss** | / | / | / | 0.9219 | 0.8496 | 0.7914 | 0.8598 | 0.3643 | 0.3400 |

The results of the first and second training cycles, i.e., are given in Table 4. Linear, ReLU, and Mish activation functions were used in 70/30, 75/25, and 80/20 train/test ratios. All models were evaluated with mAP, F1$_{score}$, Average IoU, Precision, Recall, and Average loss metrics.

In the first part of the first training cycle for the Linear activation function with 70/30, 75/25, and 80/20 train test ratios, the following results were obtained: mAP was between 59.11% and 60.18%, F1$_{score}$ was in a range between 0.47 and 0.52, Average IoU varied between 28.1% and 29.98%, the Precision score was between 0.40 and 0.44 while the Recall and Average loss in between 0.58 and 0.64 and 2.4285 and 2.6178. Progress in obtaining the final amount of Average loss can be seen in the Loss charts given in Appendix A.1.

The second part of the first training cycle for ReLU activation function evaluation metrics is changed for all tree train/test ratios. The mAP was between 94.16% and 95.12%, the F1$_{score}$ was between 0.81 and 0.87, while the Precision and Recall had similar values where the minimum value was 0.75 and the maximum was 0.93, the best Average IoU is 62.18% and the worst 58.21%. The Average loss did not change a lot and it was between 1.18 and 1.26. Charts of Average loss are presented in Appendix A.2.

In the third part of the first cycle for the Mish activation function, the following results peak values are the following: mAP value of 94.25%, F1$_{score}$ of 0.86, Average IoU 61.52%, Precision 0.81, Recall 0.93 and Average loss in the amount of 1.2489. As in the previous two parts, average loss charts can be seen in Appendix A.3.

For the fourth and fifth, parts of the second training cycle, the results of the obtained models are slightly better than in the first three parts. The maximum mAP value is 99.77% in favor of the ReLU activation function, while the minimum value is 91.27% in favor of the Linear activation function. The maximum value of the $F1_{score}$ was on several occasions 0.97 for the ReLU and Mish activation functions, which was also the peak value, while the minimum value was 0.59 for the Linear activation function. Charts of the training process can be seen in Appendix C in Appendixes C.1 and C.2.

The results of the third and fourth training cycles, i.e., 10,000 and 20,000 max batch size are given in Table 5. The Linear activation function is eliminated because of poor performance from the previous investigation. As in the last case, each training cycle of this research was conducted using the previous three train/test ratios. In the last training cycle, it is evident that the lowest average loss results were obtained compared to other training cycles. The results of mAP vary between 88.33% and 99.96%, the highest $F1_{score}$ was 1.00 while the lowest was 0.86. Both Precision and Recall ranged between 0.79 and 1. The highest score achieved for the Average IoU was 91.13%. The whole training process and average loss curve for the last two training cycles can be seen in Appendix D. The training process for the Relu activation function is shown in Appendix D.1 and for Mish in Appendix D.2.

*3.2. Discussion*

As seen from Tables 4 and 5 the entire investigation conducted in this paper can be divided into four different cycles based on max batch sizes, i.e., 6000, 8000, 10,000, and 20,000. The first cycle had 6000 batches with different activation functions and different train/test ratios. The ReLU activation function with a 75/25 ratio had the best performance in this cycle as well it had a low average loss and the best mAP percentage score.

In the second cycle of training again ReLU activation function was the best but Mish was close with its performance and metrics while in the third cycle only Mish and ReLU activation functions were evaluated. The main reason for that is because they outperformed the Linear activation function and there is no need to waste time on training the YOLOV4 algorithm with under-performed hyperparameters. In the third cycle, ReLU is not an absolute choice because the only thing that was better while using the Mish activation function is average loss. Referring to Table 5 it is visible in row Average loss that Mish is slightly better than ReLU but still, these metrics are not good enough for implementation in real situations.

In the last training cycle, the ReLu and Mish activation functions were re-evaluated for all three train test ratios. All of the metrics are higher, which contributes to the very detection of returnable packaging. A possible explanation for this is the Mish activation function has a low operating cost with a variety of properties and smooth, non-monotonic nature. It is unbounded above and bounded below that means it can avoid saturation which can cause the training to slow down. That results in more flexibility while training the algorithm and improving it [68]. The training of the algorithm was stopped at the moment when the average losses were no longer decreasing.

Looking at the fourth cycle it can be noticed that with the Mish activation function in the 75/25 train/test ratio gets outstanding results. The mAP is nearly 100%, the $F1_{score}$ is 1.00 and the Average IoU is 91.13%. The only thing that can be improved is to minimize the Average loss which is at this moment 0.3643, but there is a risk to Overfit detection algorithm model. At this moment while testing the trained detector if the thresh is put below 0.85%, the algorithm could detect wrong which is not preferable. A Linear activation function is the worst by all means while training this detection algorithm, it is not flexible as Mish and it has poor quality as shown in Table 4. By statistics, the ReLU activation function is the best but further analysis is needed to explore all possibilities while training the YOLOV4 algorithm.

Based on the values shown in Tables 4 and 5, it can be seen that the train/test ratio has slight or no influence on the classification accuracy when compared to two hyperparameters

(max batch size and activation functions). Obtained weights from the YOLOV4 model were tested with new input images that previously did not belong to the initial dataset used in this investigation. The testing was conducted using a web camera, where the percentage of reliability varied depending on the object location or angle of the camera visible from Figures 9–12.

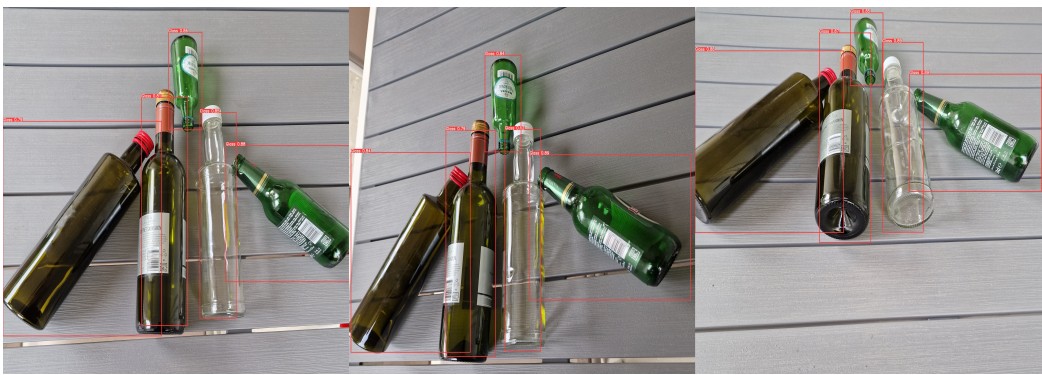

**Figure 9.** Results of YOLOV4 detection, for glass packaging using web cam.

Detection results for glass packaging are shown on Figure 9. It is visible that the YOLOV4 algorithm perfectly detects and classifies the given object. It is also visible a high coefficient of reliability for every object in the image. Plastic packaging detection results are shown in Figure 10 where it can be seen that the YOLOV4 algorithm perfectly detects and classifies plastic packaging with a high coefficient of reliability on every object on the image.

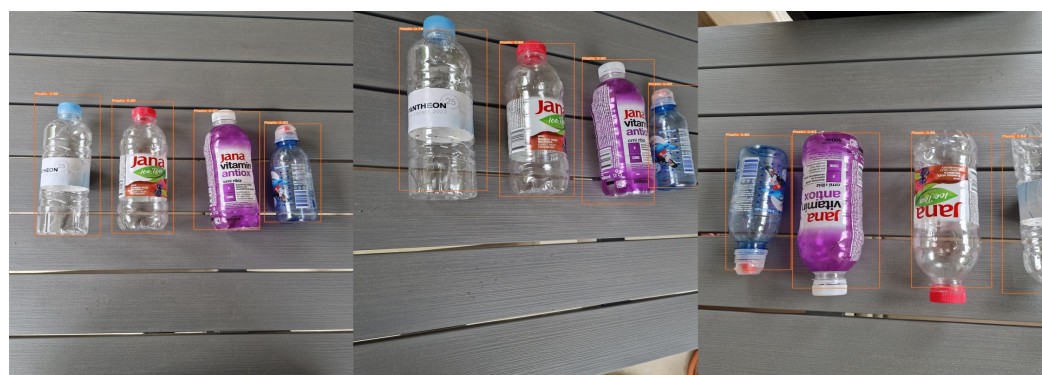

**Figure 10.** Results of YOLOV4 detection, for plastic packaging using web cam.

Aluminum packaging detection can be seen on Figure 11. As previous 2 classes, it is again visible that YOLOV4 perfectly detects and classifies the aluminum class of packaging.

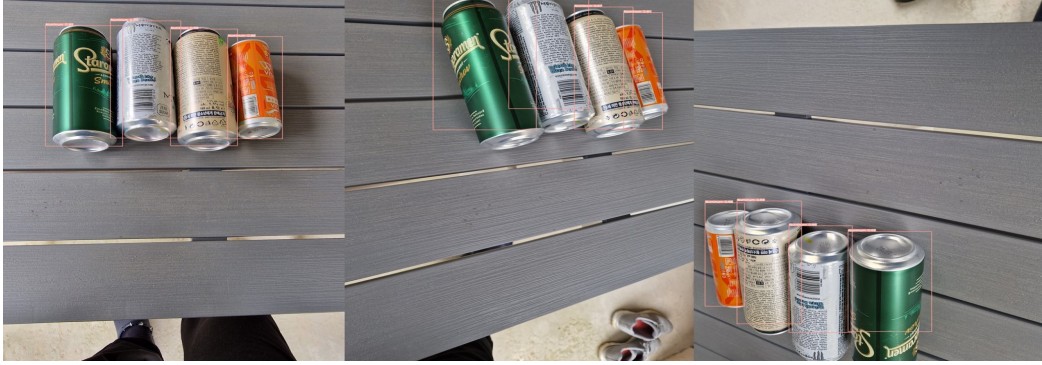

**Figure 11.** Results of YOLOV4 detection, for aluminum packaging using web cam.

The last given example of detection is a mix of all 3 classes together on the same spot as shown in Figure 12. The trained model of the YOLOV4 algorithm detects all returnable packaging with a high coefficient of reliability.

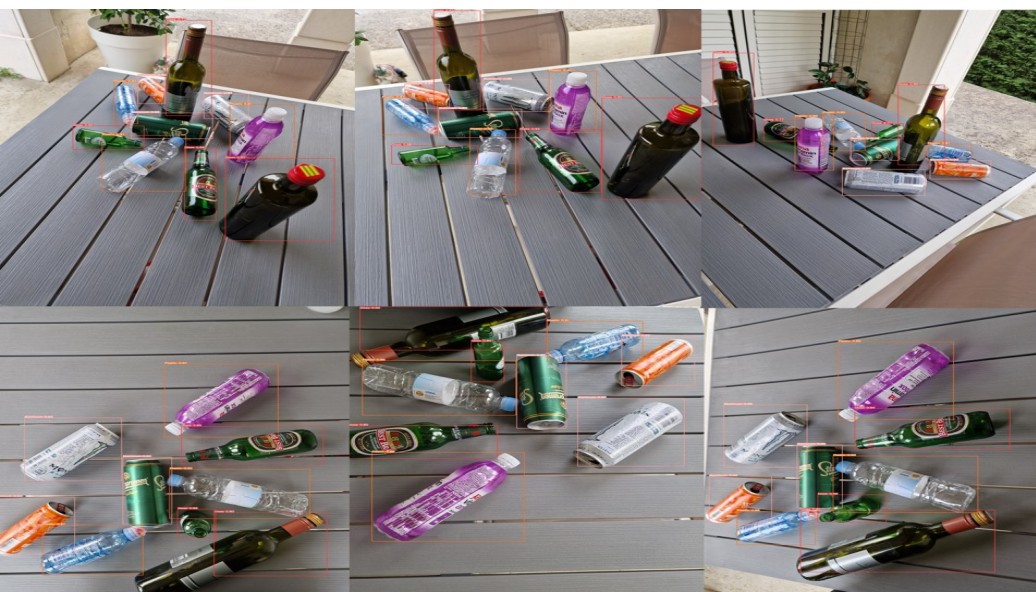

**Figure 12.** Results of YOLOV4 detection, for mixed packaging using web cam.

## 4. Conclusions

This paper presents a detection algorithm for returnable packaging recycling. The method of creating a dataset, marking bounding boxes, and what to pay attention to while compiling the dataset is shown. In addition, the method of training the desired algorithm is shown, as settings that are used with its corresponding files, and the overall configuration is shown. Toward the end of the paper, the results are presented and the differences between each training cycle are explained. Quality results can be achieved through the use of AI. For example, by using CNN, it is possible to achieve high-quality detection algorithms. In this case, YOLOV4 gives almost perfect results. So the answers to the hypotheses defined in the introduction section are:

- The YOLOV4 algorithm can achieve high classification accuracy in process of detection and classification of returnable packaging; however, the tuning process of YOLOV4 algorithm hyperparameters must be performed.
- The classification accuracy was improved by tuning just two hyperparameters: max batch size, and type of activation function used in the YOLOV4 algorithm. The investigation showed that max batch size has a greater influence than the type of activation function.
- The train/test dataset ratio did not have a notable influence on classification accuracy for a couple of reasons. The first reason is that the values of train/test ratios were the same as those ratios that are commonly used in other research papers. The lower size of the training dataset size would cause a weakly trained YOLOV4 model, while the higher size of the training dataset could cause potential overfitting of the YOLOV4 model.

This paper shows different properties of the neural networks, in this case, it can be seen that by changing the hyperparameters such as max batch size, and activation function, different results can be achieved. The quality indicators of the trained algorithm were mAP, Average loss, and Average IoU, whereby YOLOV4 at 20,000 batches in train/test ratio of 75/25 with Mish activation function shows the flexibility of data processing and gives great results. The worst example of this work is the use of the Linear activation function which shows the worst results in all training cycles. ReLU and Mish show outstanding results,

but in this case, Mish prevails because of the computational properties, training speed, and data processing flexibility which has a greater application than the ReLU or Leaky ReLU activation function. These results are shown in Tables 4 and 5, in which in the last training cycle YOLOV4 achieves results of mAP 99.96%, F1$_{score}$ 1.00 Average IoU 91.13%, and the Precision and Recall of 1.00, which is almost perfect considering the possible threat of the overfitting problem. The main contribution of this paper concerns the selection of different activation functions. The YOLOV4 detection algorithm uses the basic Mish activation function, which achieves superior properties however, the ReLU function has different computational properties than Mish. For this reason, it is not recommended to reject the ReLU activation function, because, for a different type of dataset, it is most likely that ReLU can surpass the results that are normally obtained using the default Mish activation function. Also, the importance of this paper is in devising innovative ideas that could accelerate the recycling process of recyclable materials/packaging. The problems of global climate change are visible in several aspects and future life must prevent natural disasters. Although the results are practically perfect, it is necessary to devote further research related to changing the YOLOV4 hyperparameters or YOLOV4 architecture to achieve even better and more accurate results.

**Author Contributions:** Conceptualization, Z.C.; methodology, M.G., S.B.Š. and N.A.; software, M.G.; validation, S.B.Š. and N.A.; formal analysis, N.A. and Z.C.; investigation, M.G. and S.B.Š.; resources, S.B.Š. and N.A.; data curation, M.G. and Z.C.; writing—original draft preparation, M.G. and S.B.Š.; writing—review and editing, N.A. and Z.C.; visualization, M.G.; supervision Z.C.; project administration Z.C.; funding acquisition, S.B.Š. All authors have read and agreed to the published version of the manuscript.

**Funding:** This research received no external funding.

**Institutional Review Board Statement:** Not applicable.

**Informed Consent Statement:** Not applicable.

**Data Availability Statement:** A partial set of the used dataset can be found at the link: https://www.kaggle.com/datasets/arkadiyhacks/drinking-waste-classification, (accessed on 11 October 2022).

**Acknowledgments:** This research has been (partly) supported by the CEEPUS network CIII-HR-0108, European Regional Development Fund under the grant KK.01.1.1.01.0009 (DATACROSS), project CEKOM under the grant KK.01.2.2.03.0004, Erasmus+ project WICT under the grant 2021-1-HR01-KA220-HED-000031177 and University of Rijeka scientific grant uniri-tehnic-18-275-1447.

**Conflicts of Interest:** The authors declare no conflict of interest.

## Appendix A

In this part of the Appendix section, training charts of the YOLOV4 algorithm model for 6000 max batches are attached, each sub-section represents a single activation function for all three train/test ratios.

*Appendix A.1*

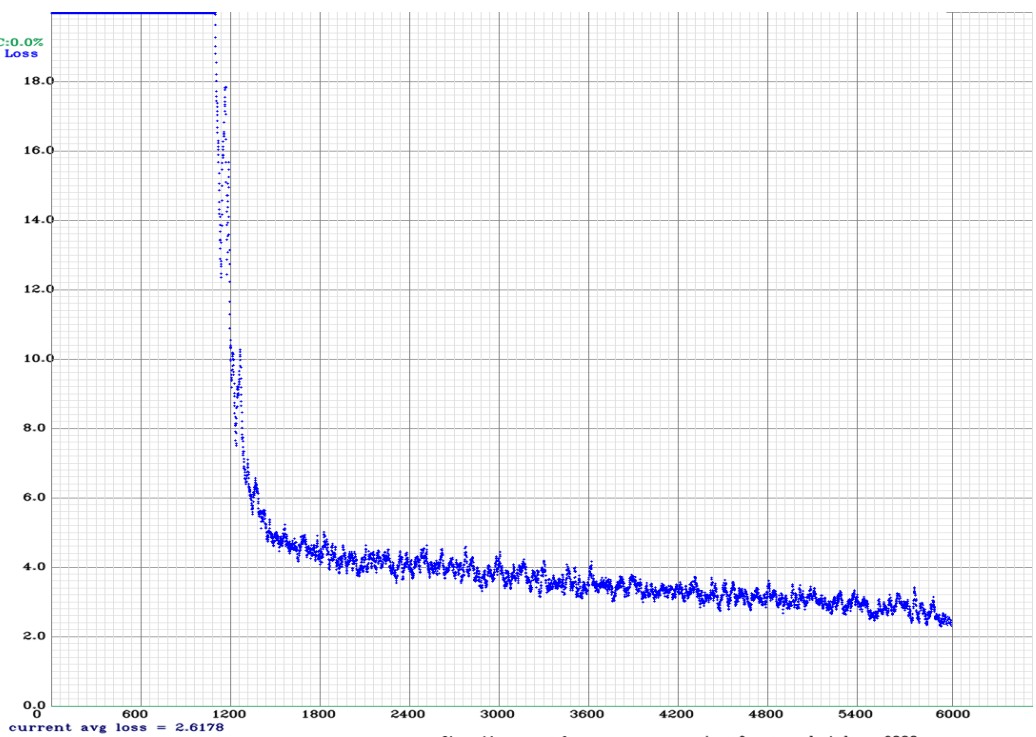

**Figure A1.** Average loss curve for 6000 max batches 70/30 train/test ratio for Linear activation function.

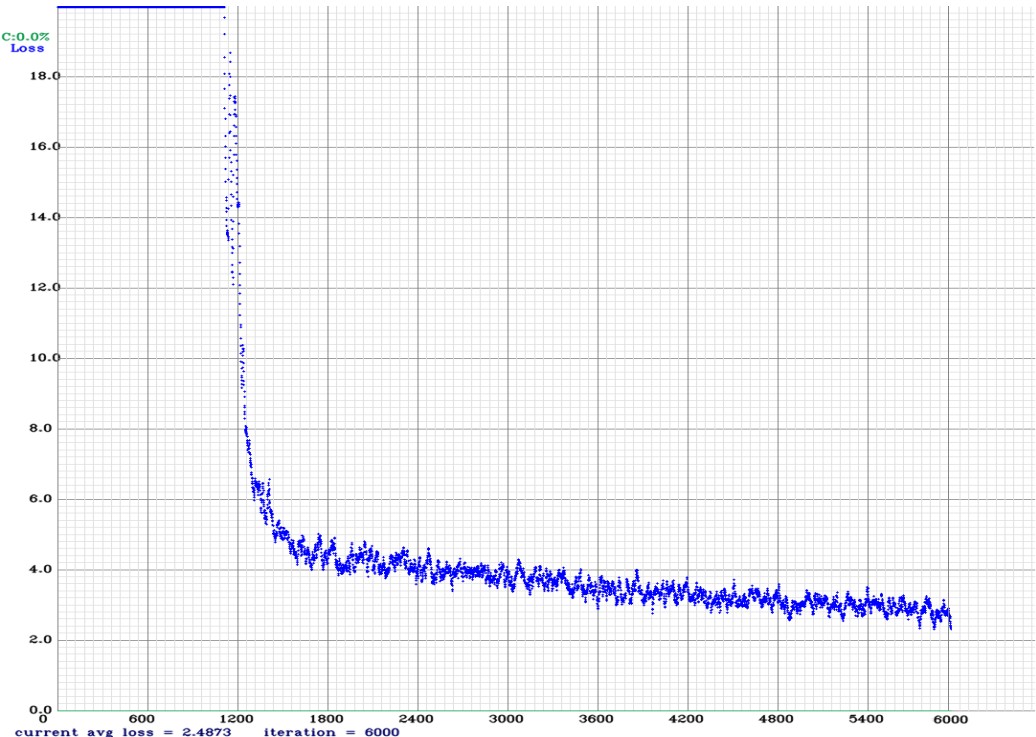

**Figure A2.** Average loss curve for 6000 max batches 75/25 train/test ratio for Linear activation function.

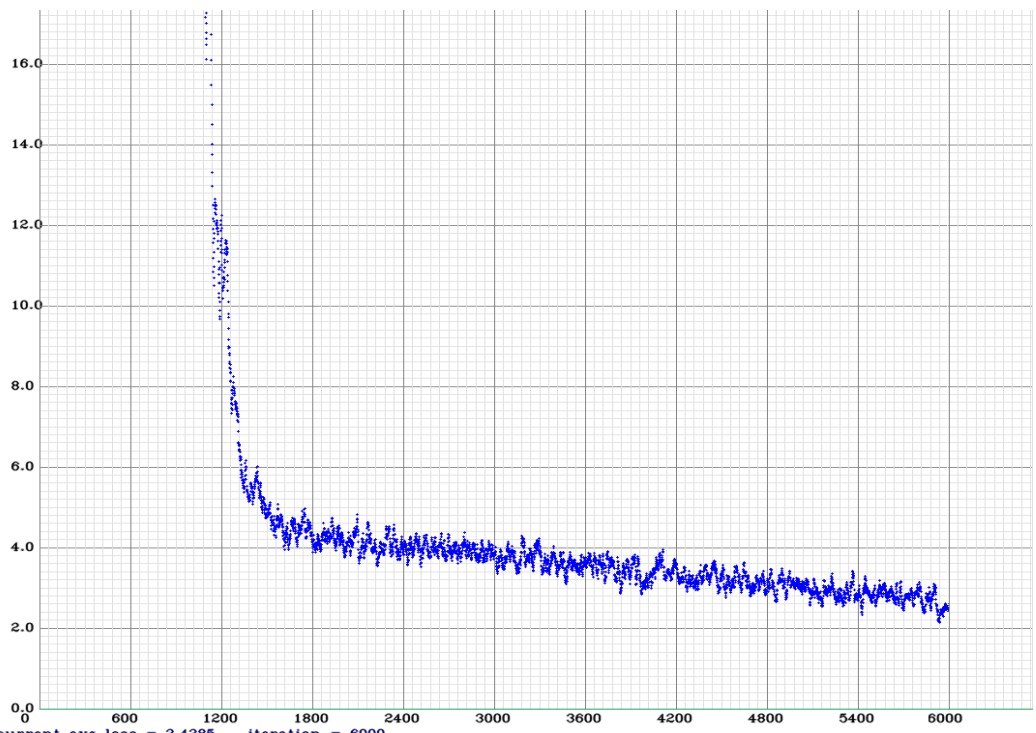

**Figure A3.** Average loss curve for 6000 max batches 80/20 train/test ratio for Linear activation function.

*Appendix A.2*

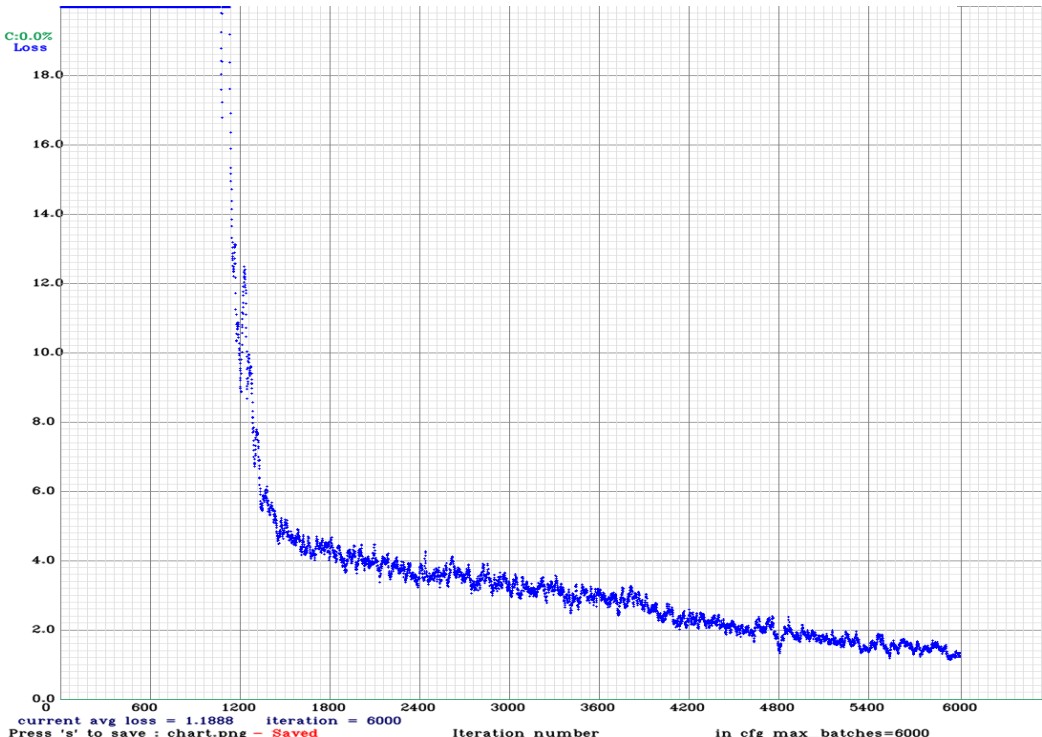

**Figure A4.** Average loss curve for 6000 max batches 70/30 train/test ratio for ReLU activation function.

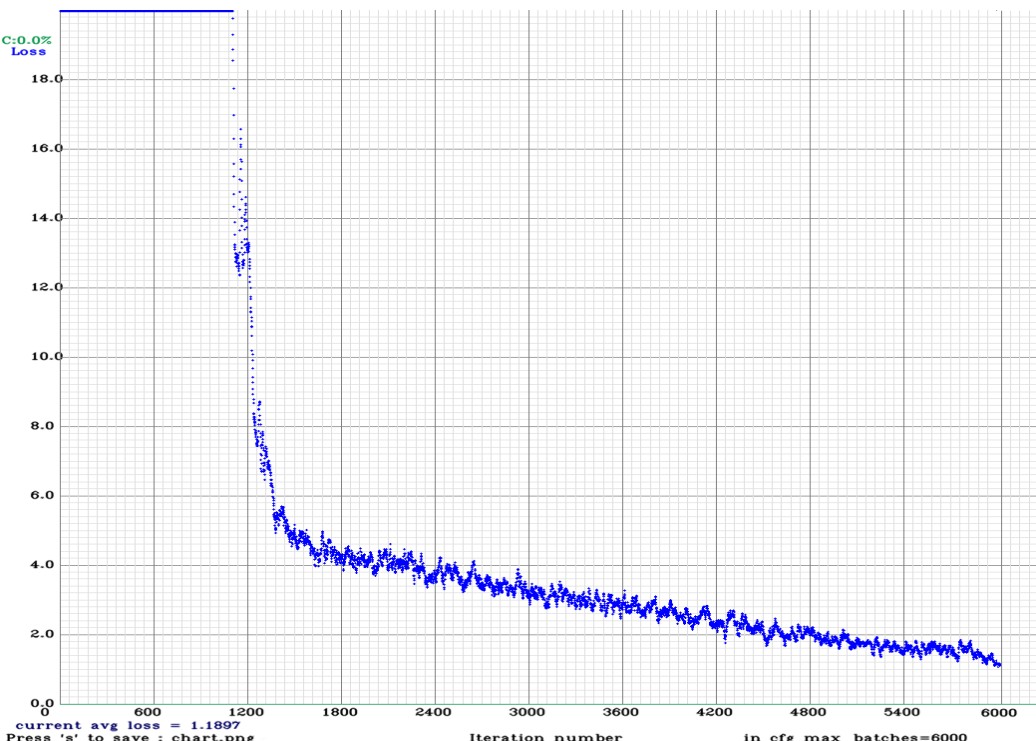

**Figure A5.** Average loss curve for 6000 max batches 75/25 train/test ratio for ReLU activation function.

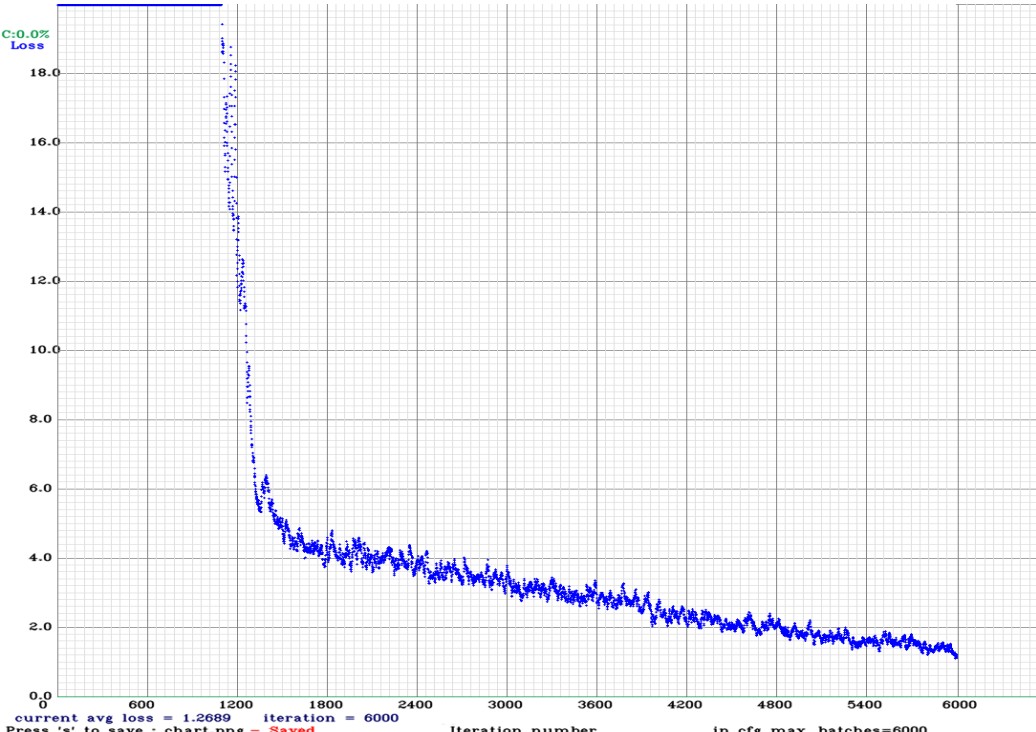

**Figure A6.** Average loss curve for 6000 max batches 80/20 train/test ratio for ReLU activation function.

*Appendix A.3*

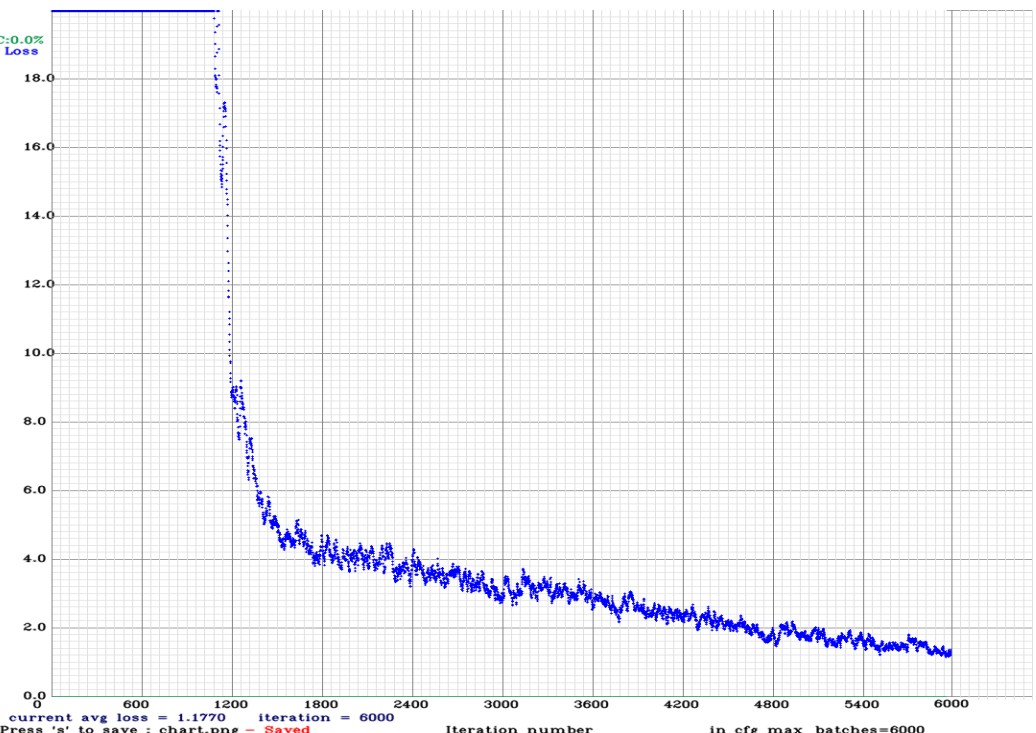

**Figure A7.** Average loss curve for 6000 max batches 70/30 train/test ratio for Mish activation function.

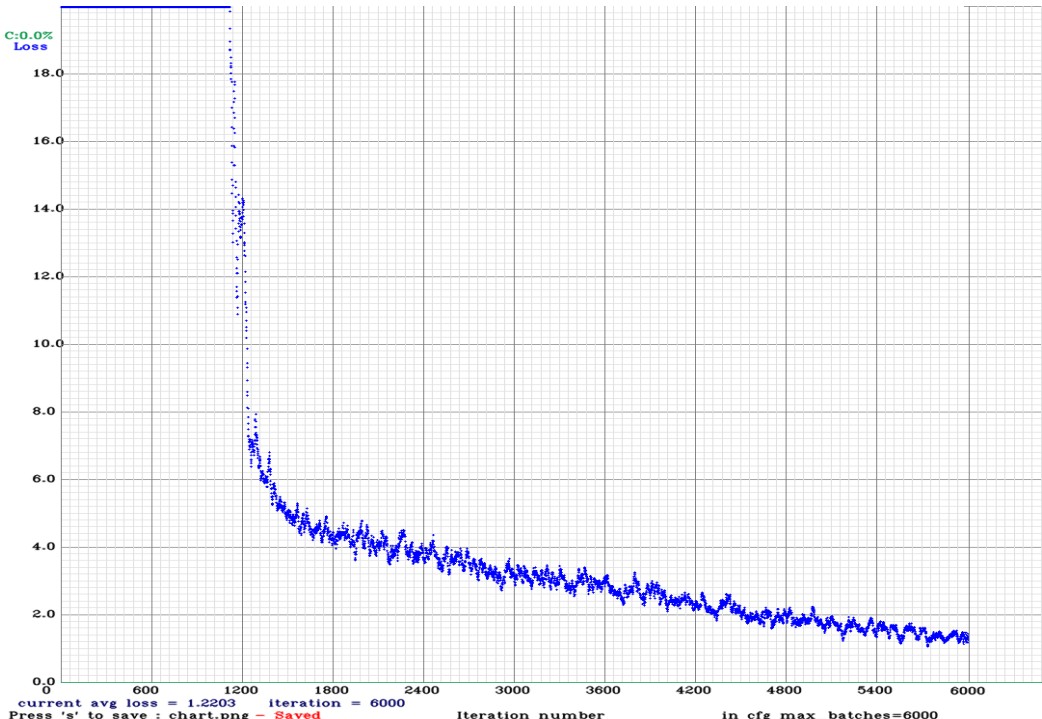

**Figure A8.** Average loss curve for 6000 max batches 75/25 train/test ratio for Mish activation function.

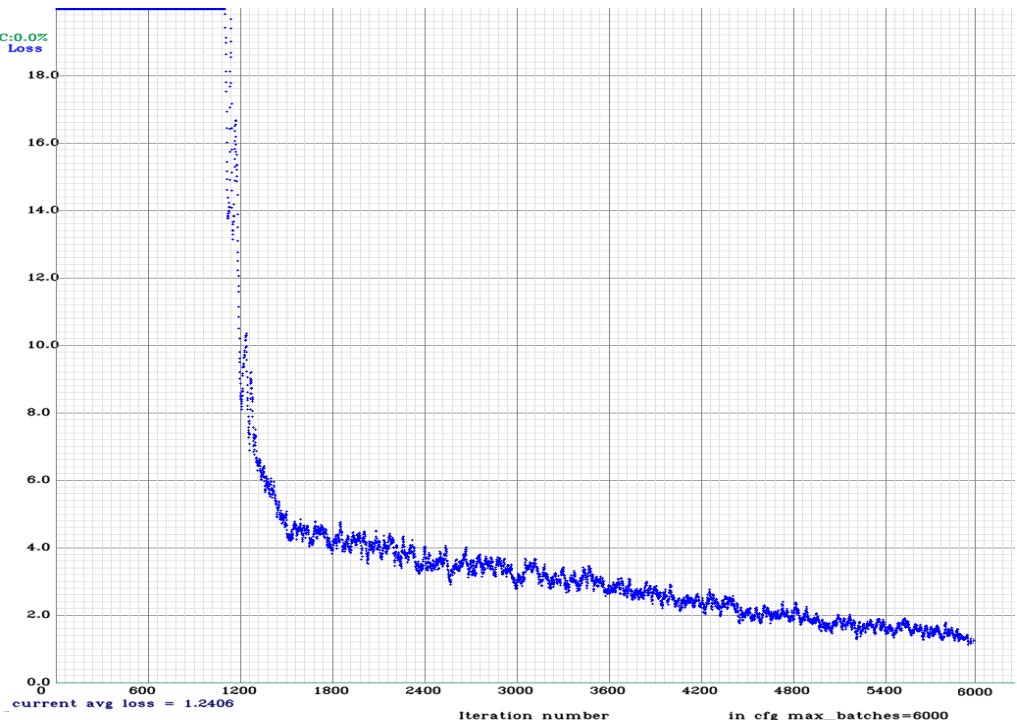

**Figure A9.** Average loss curve for 6000 max batches 80/20 train/test ratio for Mish activation function.

## Appendix B

In this part of the Appendix section, training charts of the YOLOV4 algorithm model for 8000 max batches are attached, each sub-section represents a single activation function for all three train/test ratios.

*Appendix B.1*

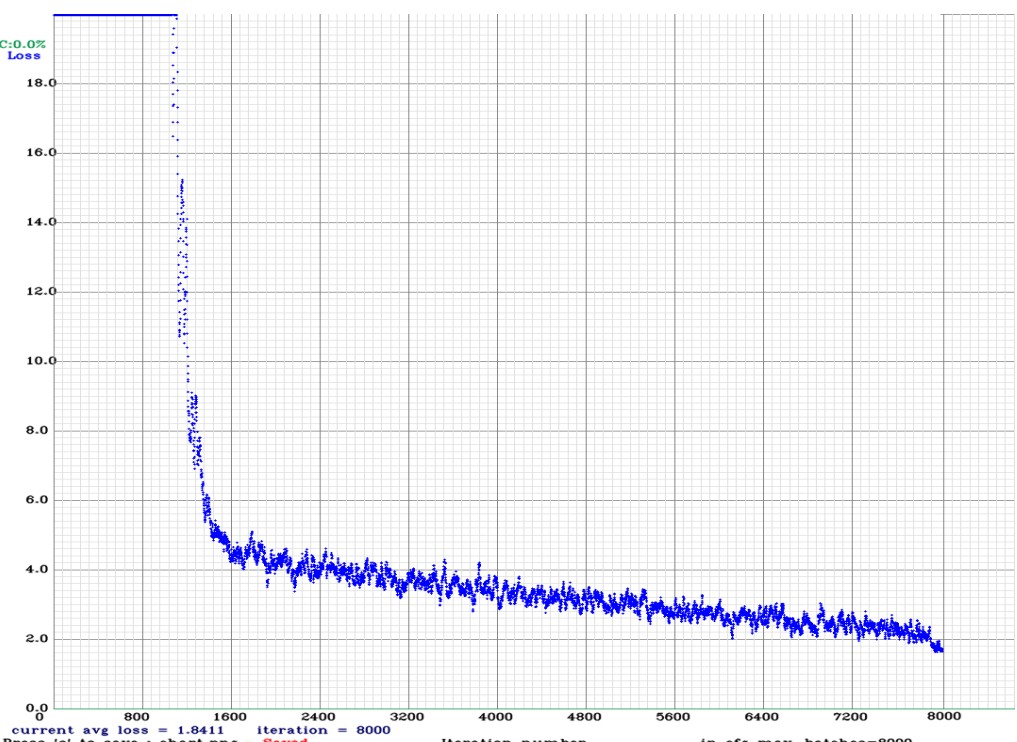

**Figure A10.** Average loss curve for 8000 max batches 70/30 train/test ratio for Linear activation function.

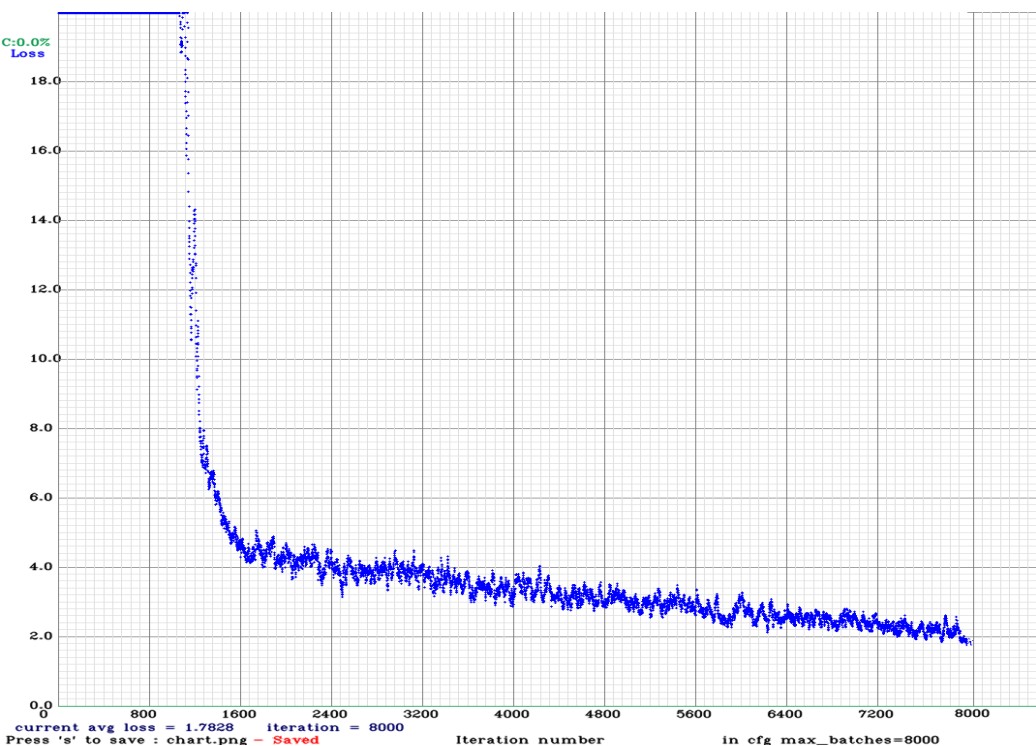

**Figure A11.** Average loss curve for 8000 max batches 75/25 train/test ratio for Linear activation function.

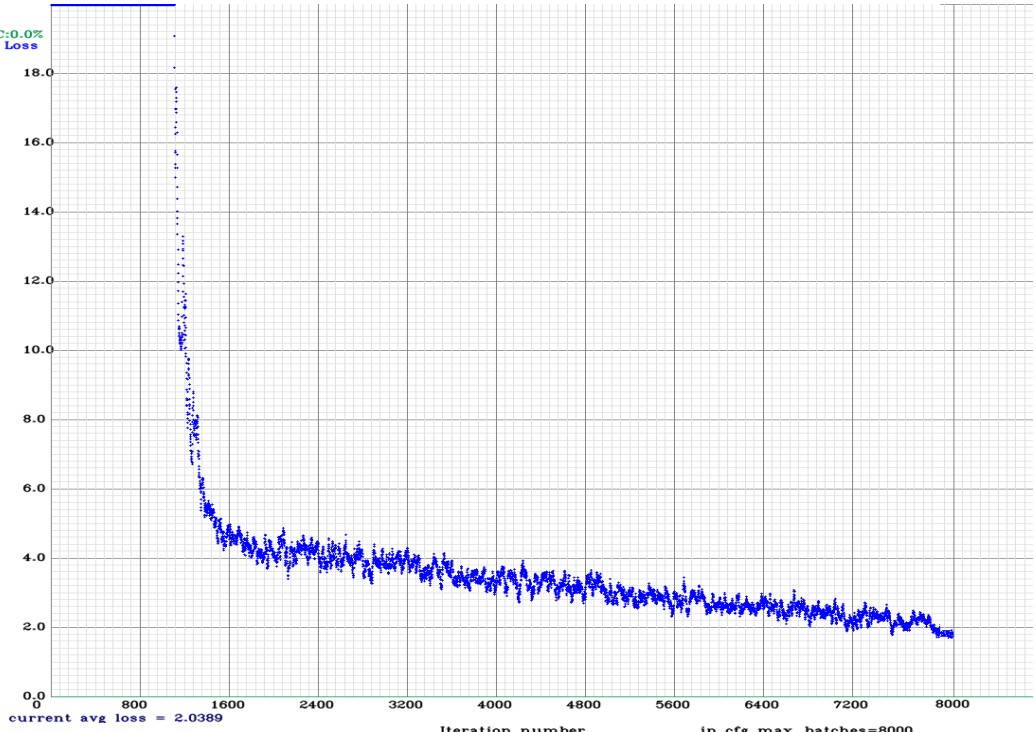

**Figure A12.** Average loss curve for 8000 max batches 80/20 train/test ratio for Linear activation function.

*Appendix B.2*

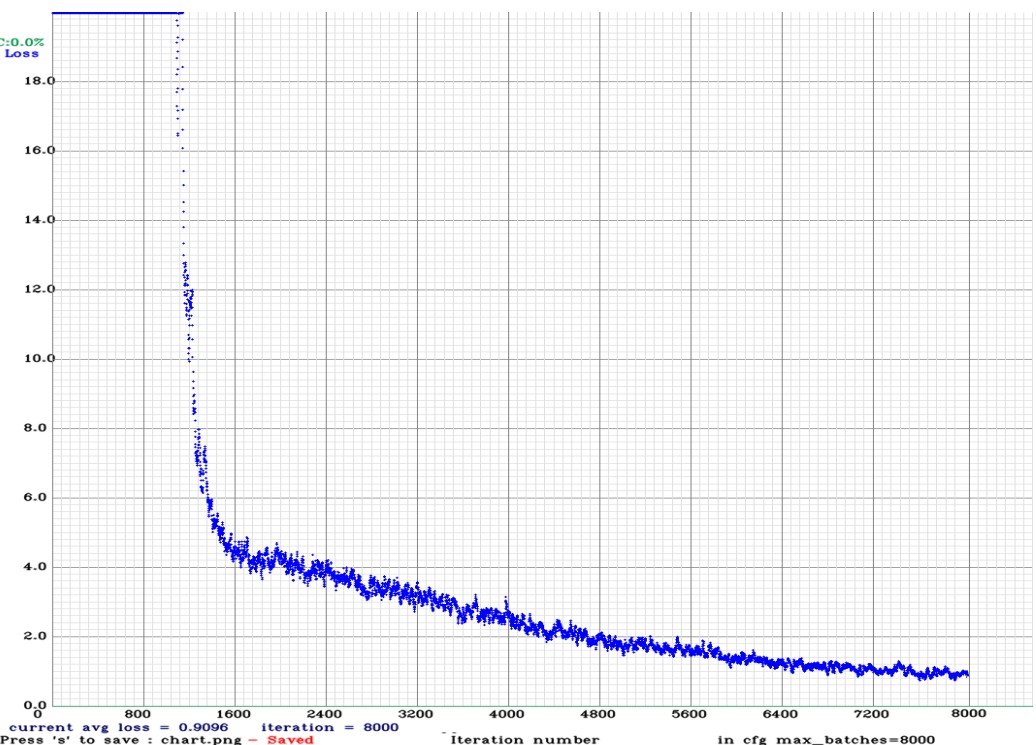

**Figure A13.** Average loss curve for 8000 max batches 70/30 train/test ratio for ReLU activation function.

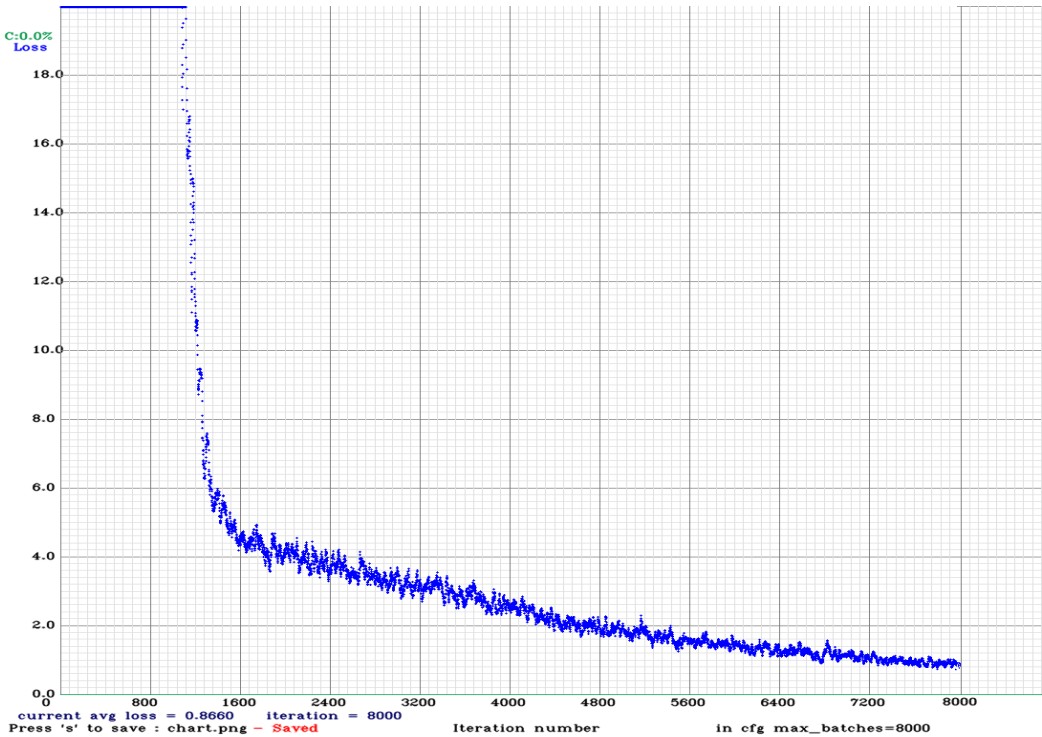

**Figure A14.** Average loss curve for 8000 max batches 75/25 train/test ratio for ReLU activation function.

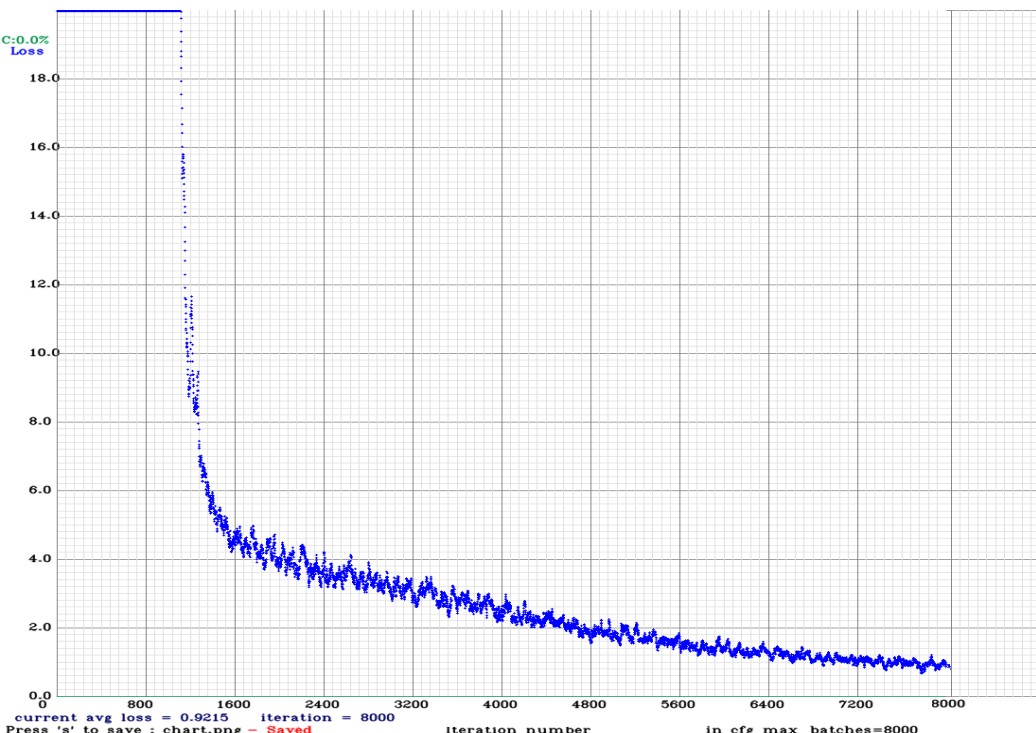

**Figure A15.** Average loss curve for 8000 max batches 80/20 train/test ratio for ReLU activation function.

*Appendix B.3*

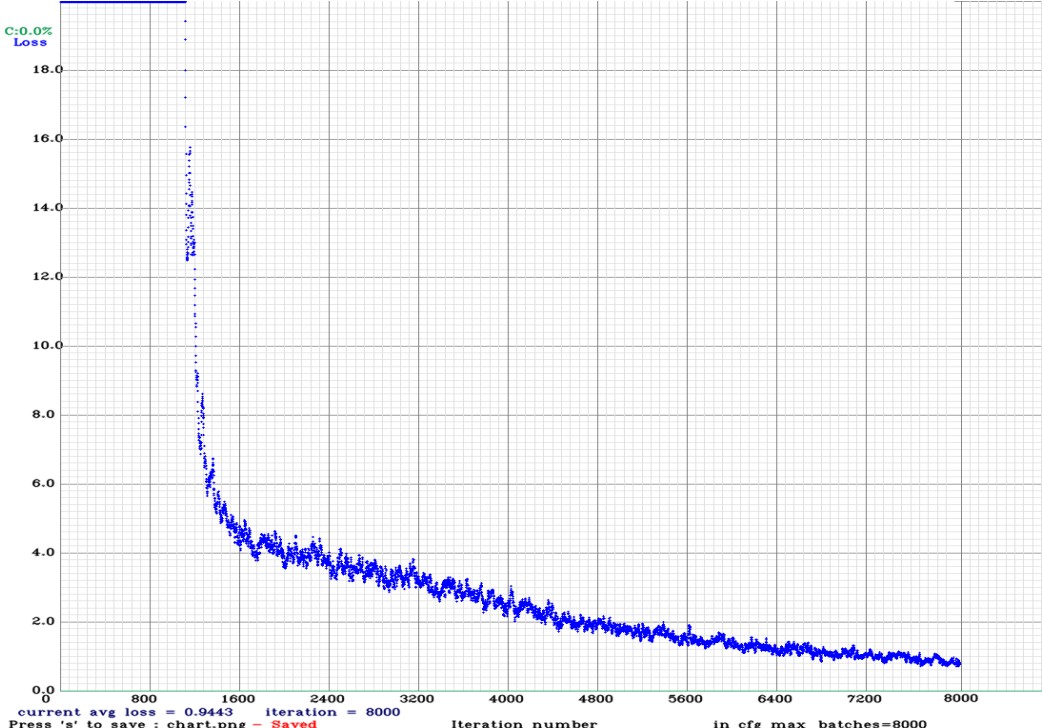

**Figure A16.** Average loss curve for 8000 max batches 70/30 train/test ratio for Mish activation function.

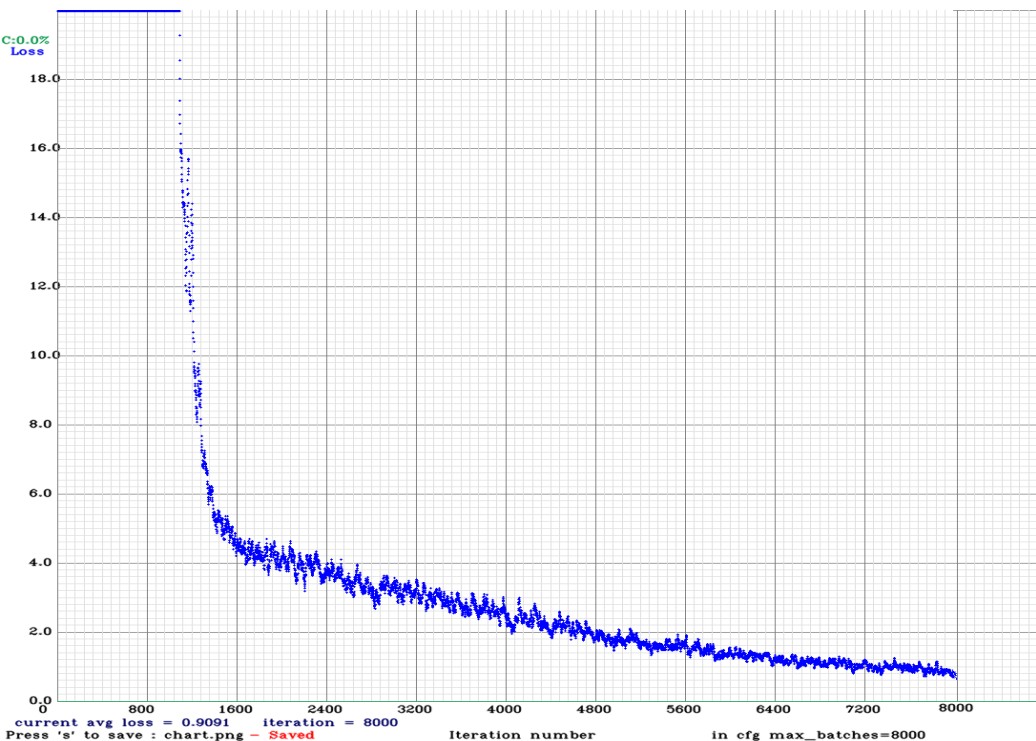

**Figure A17.** Average loss curve for 8000 max batches 75/25 train/test ratio for Mish activation function.

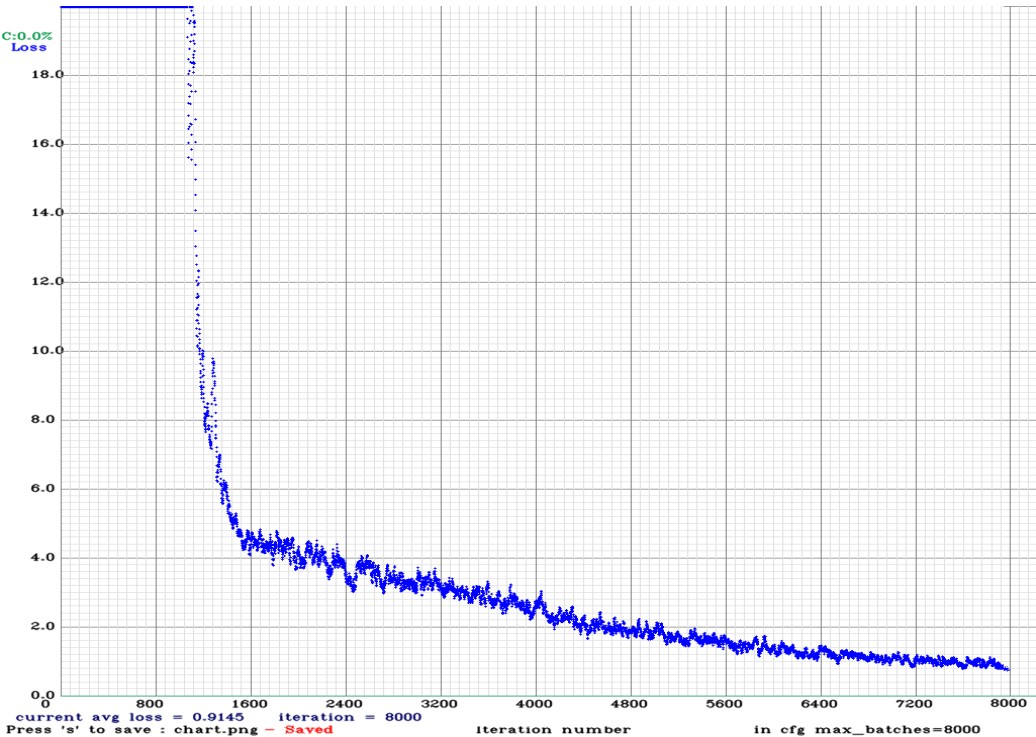

**Figure A18.** Average loss curve for 8000 max batches 80/20 train/test ratio for Mish activation function.

## Appendix C

In this part of the Appendix section, training charts of the YOLOV4 algorithm model for 10,000 max batches are attached, each sub-section represents a single activation function for all three train/test ratios.

*Appendix C.1*

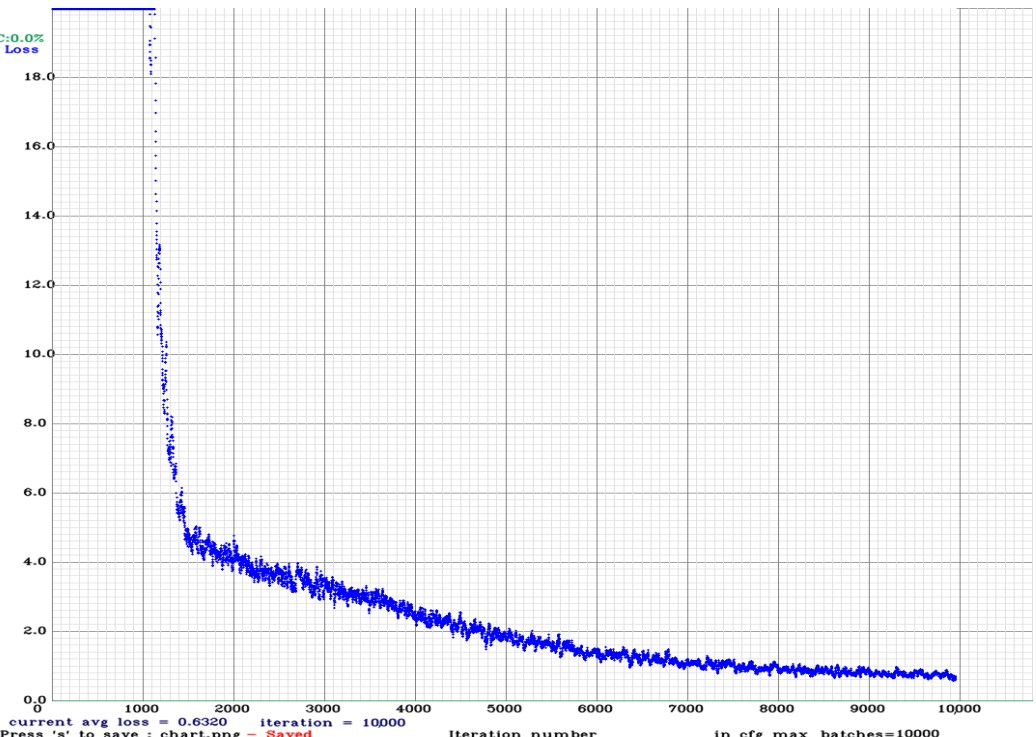

**Figure A19.** Average loss curve for 10,000 max batches 70/30 train/test ratio for ReLU activation function.

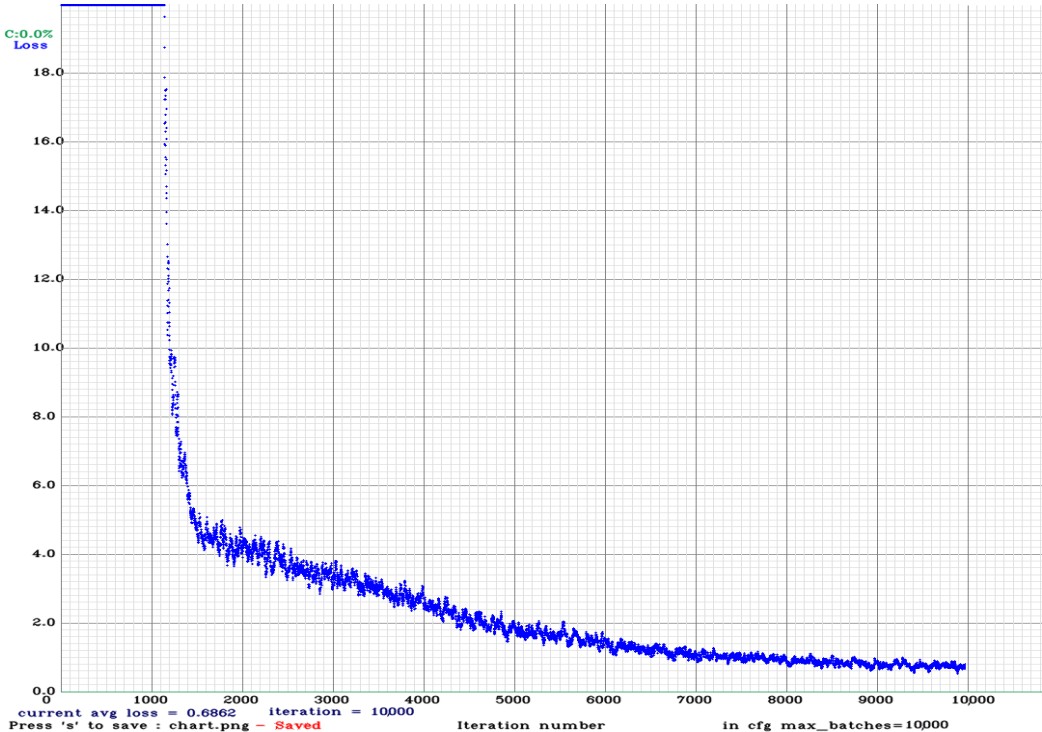

**Figure A20.** Average loss curve for 10,000 max batches 75/25 train/test ratio for ReLU activation function.

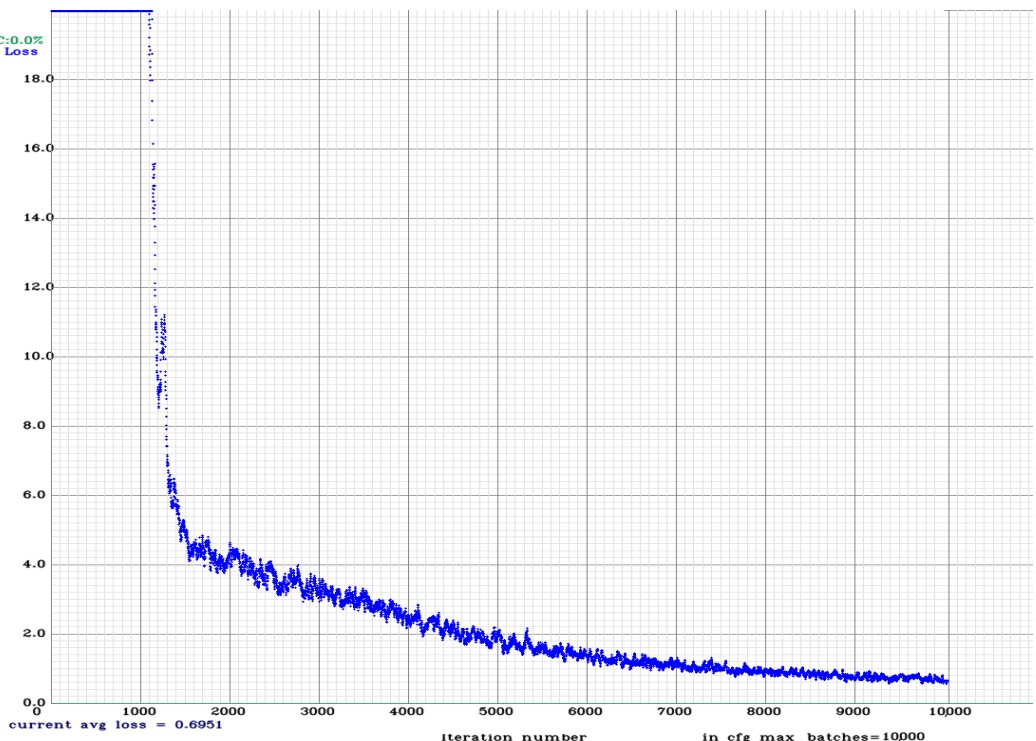

**Figure A21.** Average loss curve for 10,000 max batches 80/20 train/test ratio for ReLU activation function.

*Appendix C.2*

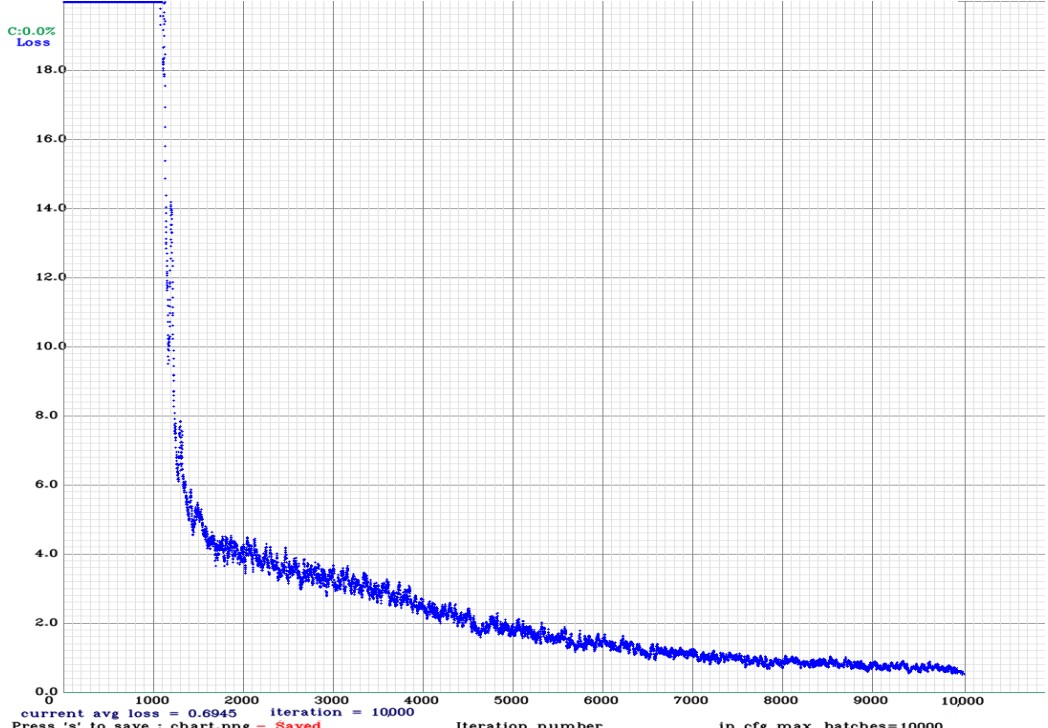

**Figure A22.** Average loss curve for 10,000 max batches 70/30 train/test ratio for Mish activation function.

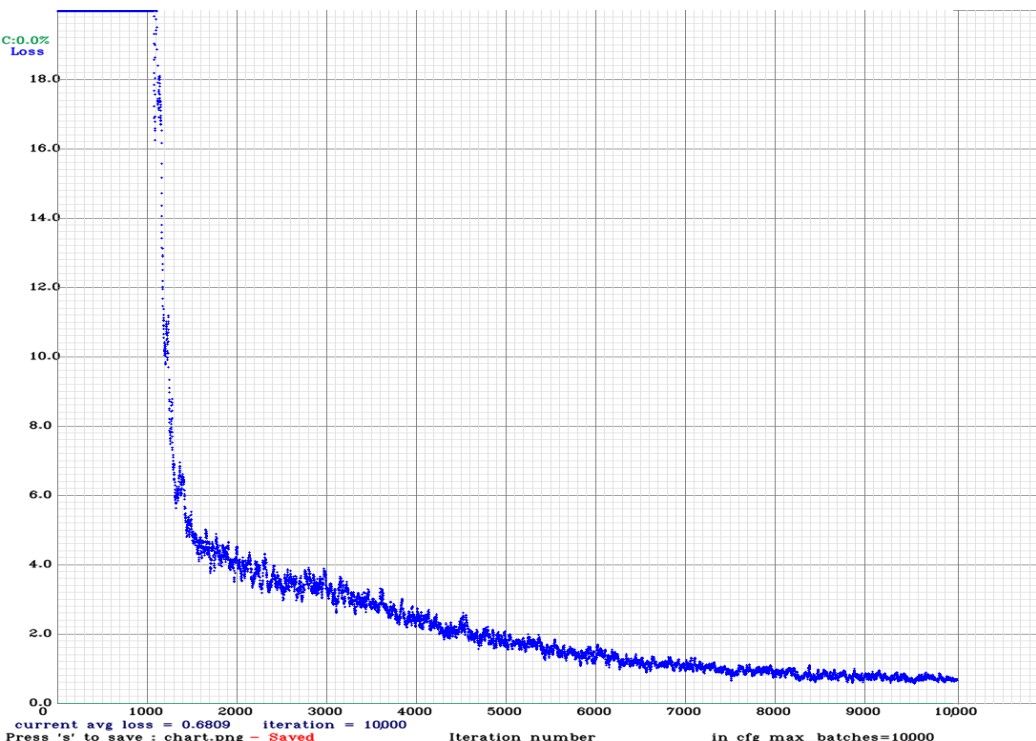

**Figure A23.** Average loss curve for 10,000 max batches 75/25 train/test ratio for Mish activation function.

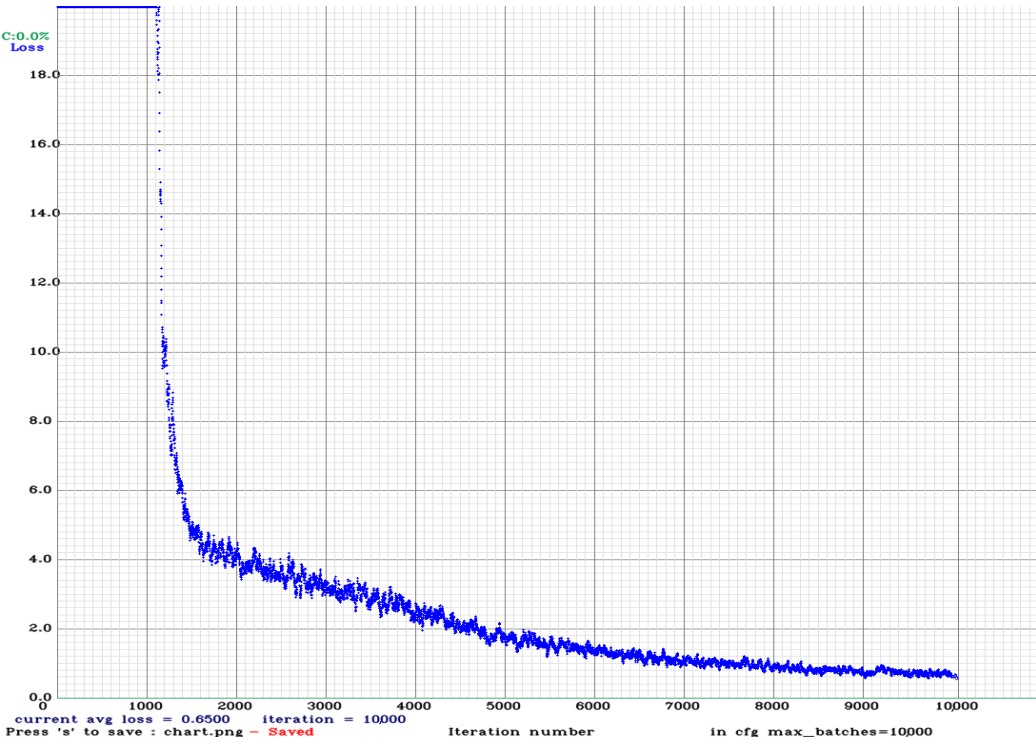

**Figure A24.** Average loss curve for 10,000 max batches 80/20 train/test ratio for Mish activation function.

## Appendix D

In this part of the Appendix section, training charts of the YOLOV4 algorithm model for 20,000 max batches are attached, each sub-section represents a single activation function for all three train/test ratios.

*Appendix D.1*

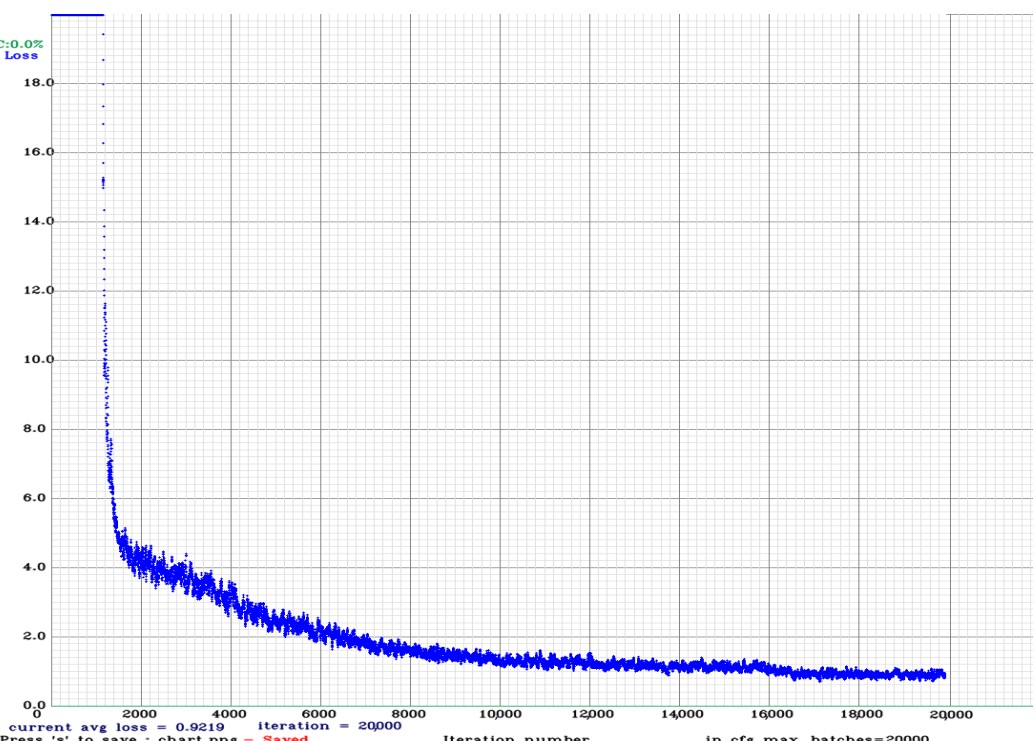

**Figure A25.** Average loss curve for 20,000 max batches 70/30 train/test ratio for ReLu activation function.

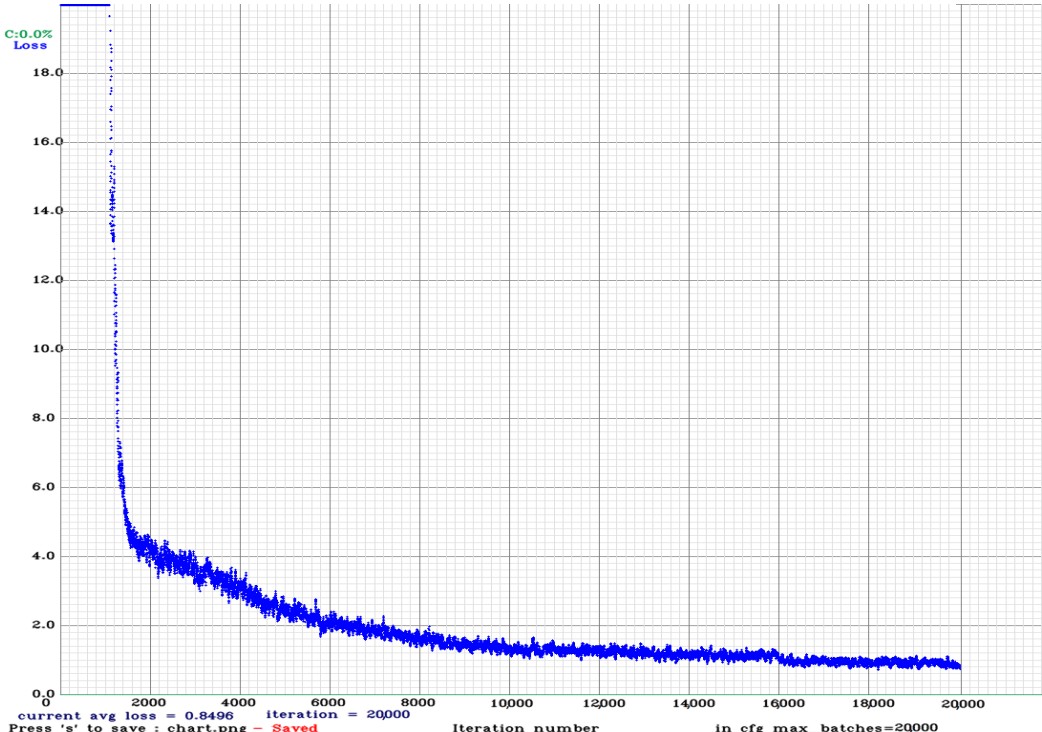

**Figure A26.** Average loss curve for 20,000 max batches 75/25 train/test ratio for ReLu activation function.

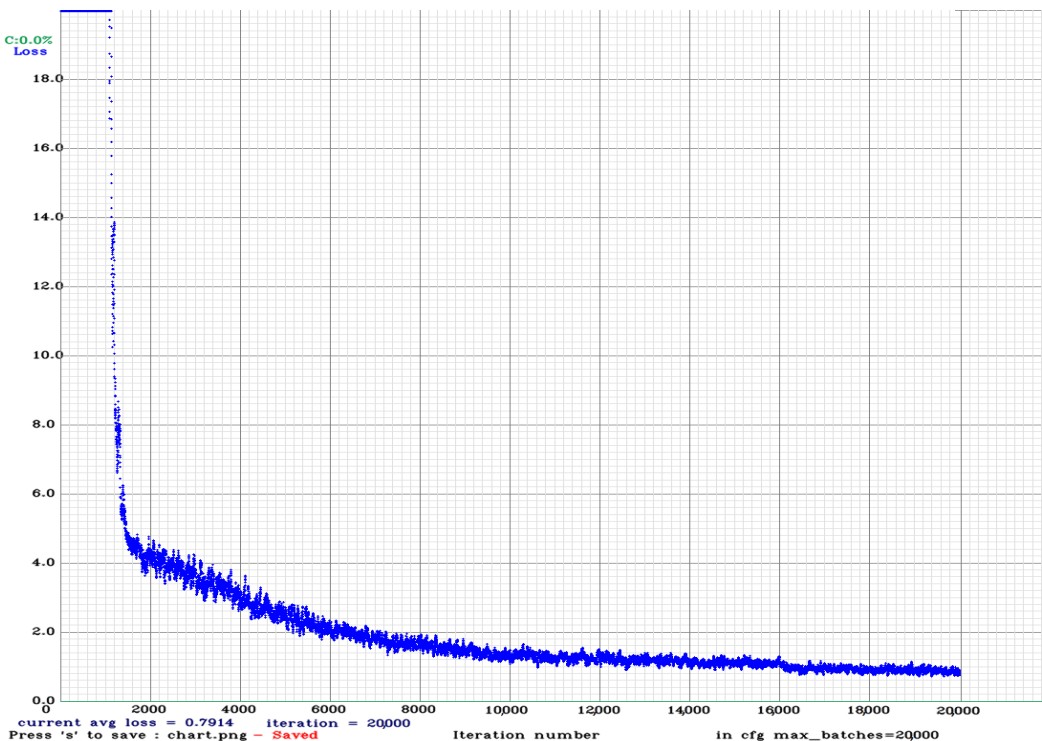

**Figure A27.** Average loss curve for 20,000 max batches 80/20 train/test ratio for ReLu activation function.

*Appendix D.2*

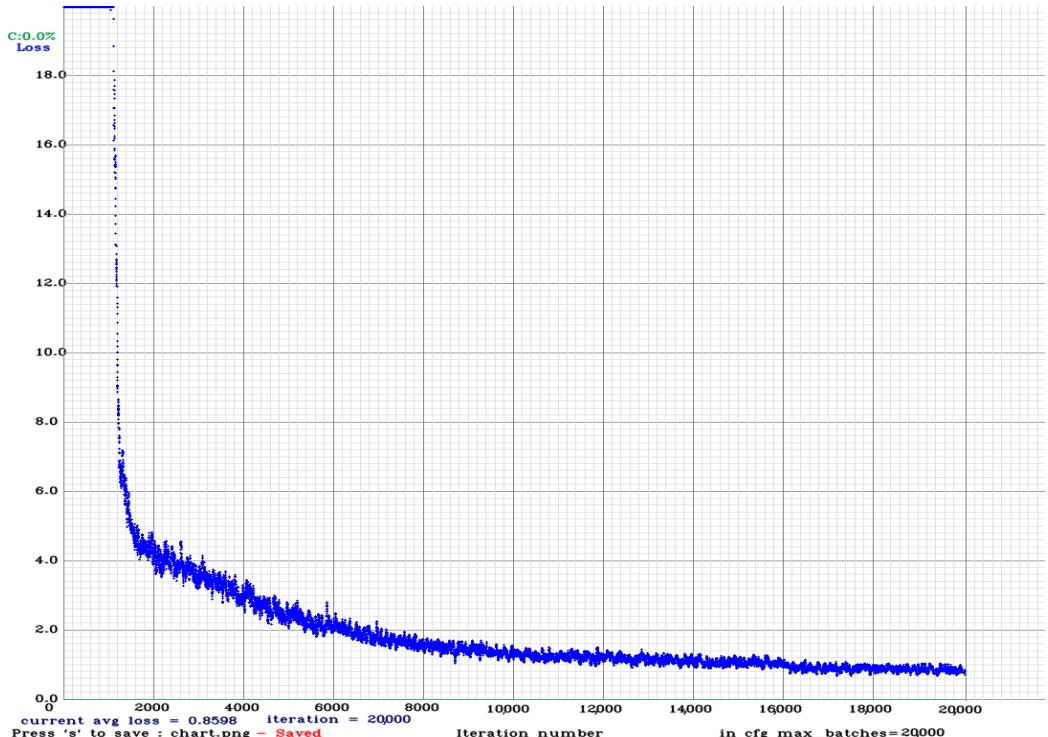

**Figure A28.** Average loss curve for 20,000 max batches 70/30 train/test ratio for Mish activation function.

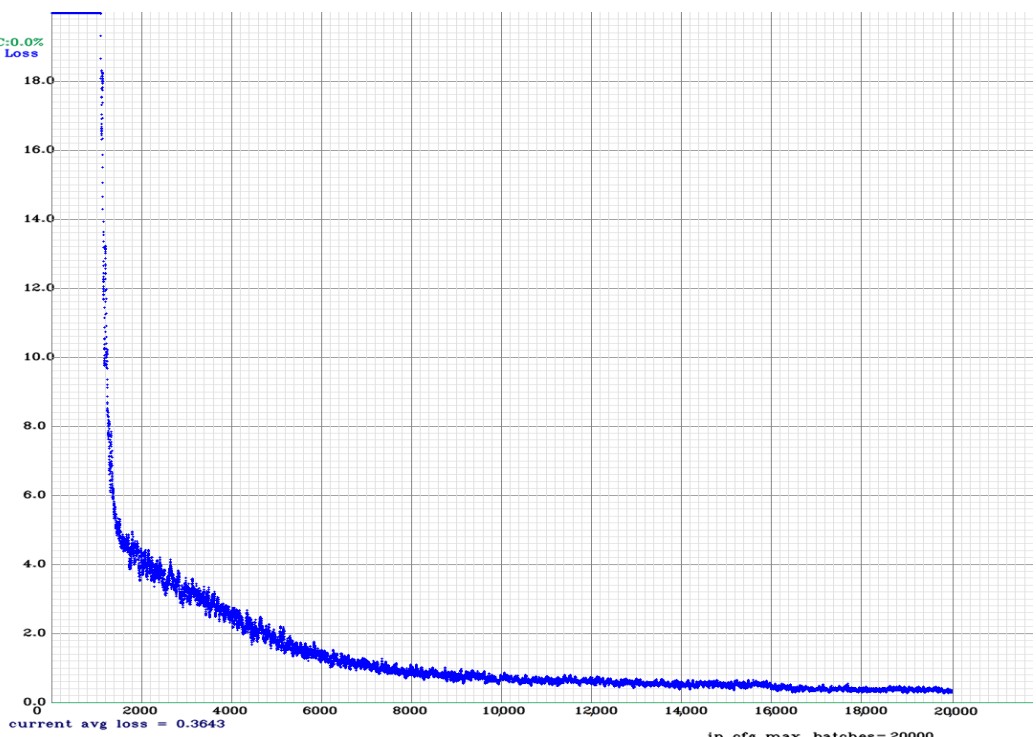

**Figure A29.** Average loss curve for 20,000 max batches 75/25 train/test ratio for Mish activation function.

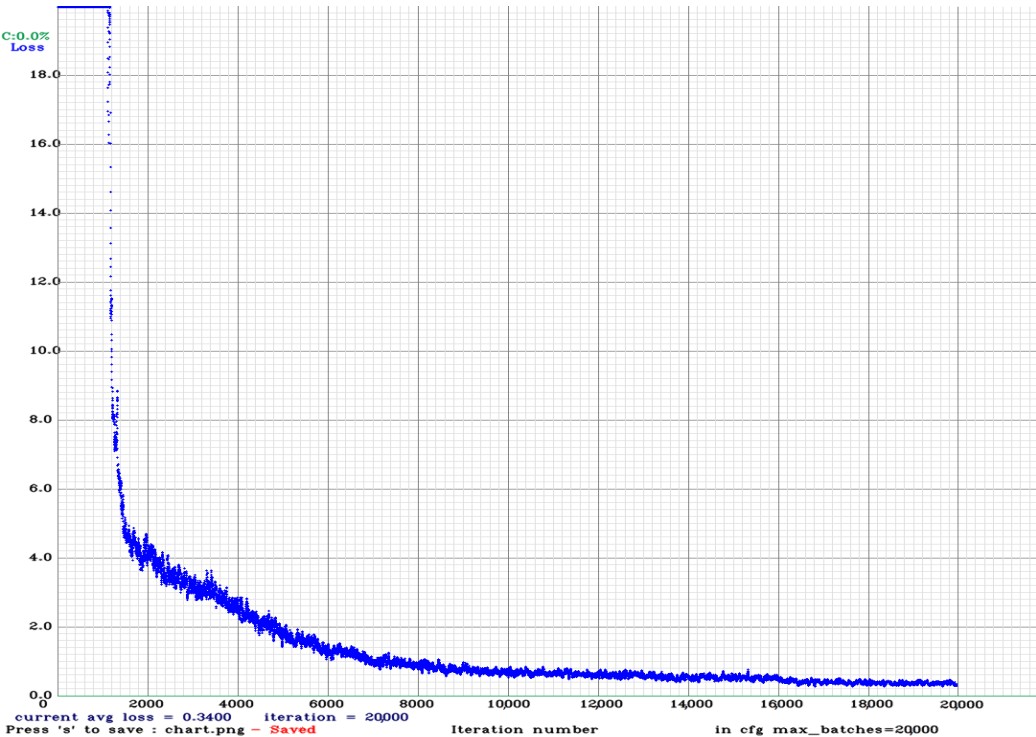

**Figure A30.** Average loss curve for 20,000 max batches 80/20 train/test ratio for Mish activation function.

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
