# Peer review of "Automated Detection and Classification of Returnable Packaging Based on YOLOV4 Algorithm"

_applsci, doi:10.3390/app122111131_

Round 1

Reviewer 1 Report

  1. The title of the paper Automated detection and classification of returnable packaging based on YOLO algorithm is very appropriate and fit for the aims of the journal.

    The abstract was well written and lucid detailing the prime aims of the study and the methodologies used to conduct the study.

    The introduction was satisfactory. However, there were couple of sentences that can be rewritten to avoid plagiarism. An instance is where the authors lifted a whole definition of waste management from [1]. 

    The methodology and materials used for the study is lucid and very appropriate. The procedure of the datasets development was very clear and appropriate.

  2.  
  3.  

    The results of each training of the YOLO algorithm with specific hyperparameters and train/test ratios were clearly presented. The model was evaluated using mean Average Precision (mAP), F1-scorePrecisionRecallAverage Intersection over Union (Average IoU) score and, Average loss. The training was conducted in four cycles i.e. 6000, 8000, 10000, and 20000 max batches with three different activation functions Mish, ReLU, and Linear (used in 6000 and 8000 max batches) is very lucid and recommendable for the test. 

  4.  
  5.  

    In the nutshell, the training and the model was seen to be very lucid, coherent and appropriate.

Author Response

Respected Reviewer,

thank you for your detailed review of our submitted manuscript. The response and answers to the questions you posed are below. Changes in the manuscript made due to your comments have been marked cyan.

R3.1. The introduction was satisfactory. However, there were couple of sentences that can be rewritten to avoid plagiarism. An instance is where the authors lifted a whole definition of waste management from [1]. 

Plagiarism occurred during the writing of the Introduction because we did not want the vital thought of the cited part to be lost. The repurposing of the text sentences will be as follows:

“Some of the disposed waste materials can last in the environment for long periods of time up to hundreds to even thousands of years for example plastics which can cause significant impacts on animals and plants.”

“Waste management is paramount for reducing and preventing the impacts of waste on nature. It can consist of several possible processes and steps which are collection, transport, treatment, and disposal of waste together with monitoring and regulation of the previously mentioned process [1].”

“For reducing the amount of waste there can be applied the waste separation methods for detection, separation, and re-useability of the returnable packaging.”

“Implementing advanced computational methods for object detection in combination with artificial intelligence (AI) algorithms can significantly improve the processes of detection and classification/separation.”

“For an intelligent and innovative way of waste separation, the key tool is object detection in combination with AI.”

“t also has had an essential role in scene understanding, which gained a high level of popularity in security, transportation, medical and military applications.”

“Applications of object detection in real-life are among others: autonomous driving [6], people detection in security [7], vehicle detection with AI in transportation [8], medical feature detection in healthcare, and in waste management [9].”

“Mao et al. [10] have used the genetic algorithm (GA) for optimization of a fully-connected-layer, DenseNet121 to improve the classification accuracy of result on ThrashNet [11] dataset which consists of 2527 images in six categories of waste (glass, paper, cardboard, plastic, metal, and trash).”

“Adedeji and Wang [12] proposed the intelligent waste material classification, which is based on ResNet-50 and support vector machines (SVM) applied to TrashNet dataset [11].

“Bobulski and Kubanek [13] performed the 48plastic waste classification using CNN on following classes: polyethylene terephthalate, 49high-density polyethylene, polypropylene and polystyrene. Authors in the research created 50a unique dataset, they also investigated the different configurations of CNN (15 and 23- 51layer network), diverse image resolutions (....“

“In this investigation, the highest classification accuracy achieved was 92.6%”

“The highest accuracy score was 95.35% and it was achieved in ResNet50. On the original dataset Altikat et al. [17] applied the four and five-layer deep CNN for classifying the…”

“In the last few years, You-Only-Look-Once (YOLO) algorithm has been increasingly used in the detection and classification of non-biodegradable waste. Authors Aishwarya et al. [18] applied the YOLO algorithm for the detection of non-biodegradable waste from the bins. Classification accuracy was faulty and imprecise with the results of 75% 70for metal, 65% for plastic, and 60% for glass.

“here the classification accuracy of recyclables using YOLO was 91% in an optimal computing environment and 75% when implemented on Raspberry Pi. Lin [20] used the YOLO-Green object detection algorithm for detecting trash in real-time.”

We have used cyan highlight to note the changes made to the manuscript due to your comments. We hope you will be satisfied with our answers to the questions posed.

Kindest regards,
The Authors

Reviewer 2 Report

The authors evaluate the performance of the YOLO algorithm in detecting returnable packaging. The methods and results are well structured. Still, I have the following questions.

1. The type of items in the datasets are relatively small, which are Plastic, Glass, and Aluminum. Are there any other types of objects belonging to the returnable objects?  

2. The performance in real applications is vital. Therefore, conducting more experiments on real, complex scenarios would be better. 

Author Response

Respected Reviewer,

We thank you for your time and effort in reviewing our manuscript. The answers to the comments are posed below. In the manuscript, all changes are marked in yellow.

R1.1. The type of items in the datasets are relatively small, which are Plastic, Glass, and Aluminum. Are there any other types of objects belonging to the returnable objects? 

The following three materials are acceptable returnable packaging in the Republic of Croatia: aluminum cans, plastic bottles, and glass bottles. Therefore, there are only three classes of materials used in the research. So the answer to your question would be no, there are no other materials related to returnable packaging. Regarding recycling, there are other types of materials that can be recycled, however, that was not the initial goal of this research.

R1.2. The performance in real applications is vital. Therefore, conducting more experiments on real, complex scenarios would be better.

We agree with your remark and corrections have been applied to the paper. Added Figures are in the Result section.

We hope you will be satisfied with our answers to the questions posed. We have marked the changes made to the manuscript due to your comments using yellow highlight.

Kindest regards,
The Authors

Reviewer 3 Report

Summary:

This paper proposed to apply the YOLO model to detect and classify returnable packaging images. The authors created partial new datasets that combined existing images with images downloaded from Kaggle. The authors then performed parameter tuning to get the highest performance of the created datasets.

Strengths:

1. The details of the dataset creation and labeling processes are clear.

2. The training of created datasets is well presented.

Weaknesses:

1. The method contributions of this paper are limited. This method only explored the YOLO algorithm, and it just applied YOLO in returnable packaging images.

2. The authors did not compare with SOTA methods, and it did not know which version of YOLO was used. Currently, there are several versions of YOLO available, such as YOLO V3, V5, ..., and V7. In addition, other methods should be compared, like MaskRCNN, FastRCNN, G-RCNN, etc.

3. There is redundant information from Figure 10 to Figure 17. It is better to put them in the supplementary material.

4. Another significant weakness of the paper is that there is a fundamental flaw in the experiments. The experiments are not completed, and we cannot draw the conclusion as in the paper from current experiments in Table 3 and Table 13. For example, to explore which activation function is the best, all max batches and train/test ratio should be the same. If the authors want to explore which split is the best, all max batches and activation functions should be the same. If the max batches is 600, then the following should be a complete experiment:

No.

max batches

activation function

train/test ratio 

1

6000

ReLU 

75/25

2

6000

Linear

75/25

3

6000

Mish

75/25

4

6000

ReLU 

70/30

5

6000

Linear

70/30

6

6000

Mish

70/30

7

6000

ReLU 

80/20

8

6000

Linear

80/20

9

6000

Mish

80/20

Similarly, for max batches = 8000, 10000, and 20000, the above experiments should be added. Then, we can draw a conclusion about which batch number, which activation function, and which split is the best for the created dataset.

Author Response

Respected Reviewer,

thank you very much for your review of our manuscript. We have tried our best to respond to the issues you have noted in your manuscript. Please find our responses below. Changes in the manuscript made due to your comments have been marked yellow.

R2.1. The method contributions of this paper are limited. This method only explored the YOLO algorithm, and it just applied YOLO in returnable packaging images

Given that it is about the detection and classification of returnable packaging, the application of this type of algorithm and the obtained model is in dynamic and fast systems, for example, a conveyor belt for collecting returnable packaging. Based on multiple detailed research, the YOLO algorithm was taken into account due to its performance, speed, and accuracy. Changes were made in the Introduction section of the paper and marked in quotation marks below.

“In this paper, the optimal performance of the YOLOV4 algorithm is examined, for example, Kim et al. [22] in his article applied YOLOV4 algorithm and achieves the most optimal ratio of mAP and frames per second (FPS) in real-time vehicle detection, while Faster-RCNN had the worst performance, i.e. it was the slowest and least accurate compared to other algorithms. Single Shot Detector (SSD) was the fastest but it had significantly lower Precision than Faster-RCNN and YOLOV4. In terms of object localization, according to Ye et al. [21], it is evident that the YOLO detection algorithm compared to RCNN, RRPN, RetinaNet-H, RetinaNet-R, R-3Det, RCI & RC2, RPN, RRD and, ROI-Transformer, was the fastest with 60 FPS value compared to others. Robotic systems also have applications and impacts on reducing the amount of garbage in the environment. AI-controlled Outdoor Autonomous Trash-Collecting Robot is used for underwater garbage collection [23]. Several object detection algorithms such as Mask-RCNN YOLOV4 and YOLOV4-tiny were compared and tested for underwater waste detection and classification. Results show that the YOLO algorithm had the highest mAP and detection speed compared to other used algorithms. The final but most important and main reason, besides the robustness and high rate of Precision and detection speed, is the size of the weights generated by the YOLO detection algorithm. Tian et al. [24] tested several AI algorithms in which 4SP-YOLOV4 compared to Faster R-CNN, SSD, etc. has had a much smaller weight file size which can lead up to a possibility of a boarder range of the model implementation to printed circuit board-based microcontrollers such as Raspberry Pi, CaffeLatte, etc”

R2.2. The authors did not compare with SOTA methods, and they did not know which version of YOLO was used. Currently, there are several versions of YOLO available, such as YOLO V3, V5, ..., and V7. In addition, other methods should be compared, like MaskRCNN, FastRCNN, G-RCNN, etc.

In the article, the YoloV4 version is defined under the keywords section, the correction was made according to the given comment. Every term associated with the article from YOLO was redefined as YOLOV4. Regarding the comparison of MaskRCNN, FastRCNN, G-RCNN, and other algorithms with the YOLOV4 algorithm. The YOLOV4 algorithm was chosen because of the needed requirement. In this case, optimal speed and accuracy of detection and classification of returnable packaging were required for possible implementation of the resulting model in a real environment. Given that it is about fast systems, i.e. the original purpose is fast systems as a conveyor belt in a recycling plant, it is important that the model quickly recognizes the object and makes the most accurate classification possible. During the previous research and additional corrections below in the paper, YOLO perfectly meets the stated requirements.

“In this paper, the optimal performance of the YOLOV4 algorithm is examined, for example in the article [22] the mentioned algorithm has the most optimal ratio of mAP and frames per second (FPS) in real-time vehicle detection, while Faster-RCNN had the worst performance, i.e. it was the slowest and least accurate comparing to other algorithms. Single Shot Detector (SSD) was the fastest but it had significantly lower precision than Faster-RCNN and YOLOV4. In terms of object localization, according to paper [21], it is evident that the YOLO detection algorithm compared to RCNN, RRPN, RetinaNet-H, RetinaNet-R, R-3Det, RCI & RC2, RPN, RRD and, ROI-Transformer, was the fastest with 60 FPS value compared to others. Robotic systems also have applications and impacts on reducing the amount of garbage in the environment. AI-controlled Outdoor Autonomous Trash-Collecting Robot is used for underwater garbage collection [23]. Several object detection algorithms such as Mask- RCNN YOLOV4 and YOLOV4-tiny were compared and tested for underwater waste detection and classification. Results show that the YOLO algorithm had the highest mAP and detection speed compared to other used algorithms. The final but most important and main reason, besides the robustness and high rate of precision and detection speed, is the size of the weights generated by the YOLO detection algorithm. The article [24] tested several AI algorithms in which 4SP-YOLOV4 compared to Faster R-CNN, SSD, etc. had a much smaller weight file size which opens up the possibility of a boarder range of the model implementation to printed circuit board-based microcontrollers such as Raspberry Pi, CaffeLatte, etc”

R2.3. There is redundant information from Figure 10 to Figure 17. It is better to put them in the supplementary material. 

Figures of the chart loss are moved to the Appendix section and are divided according to the training process of the YOLOV4 algorithm. The Appendix section is divided into four categories A, B, C and D, which indicate training cycles of 6000, 8000, 10000, and 20000 iterations, and three subcategories 1, 2, and 3, which indicate training for Linear, ReLU, and Mish activation functions.

R2.4. Another significant weakness of the paper is that there is a fundamental flaw in the experiments. The experiments are not completed, and we cannot draw the conclusion as in the paper from current experiments in Table 3 and Table 13. For example, to explore which activation function is the best, all max batches and train/test ratio should be the same. If the authors want to explore which split is the best, all max batches and activation functions should be the same. If the max batches is 6000, then the following should be a complete experiment. 

Remarks were taken into account and additional training on missing models was done. In doing so, the following changes were made. Table 3 has been modified, and information in the Results and discussion section has also been added. In addition, the previous Table of results on page 11 has been reformulated in Table 4 and Table 5 with added missing values ​​for each model, which makes the total number of 30 models. In addition, each training cycle and part of each cycle are described in the text for Tables 4 and 5. In the Discussion section, there is a description of the selection of the best model. The following text was added to the paper's Results and discussion section:

“and the key points during the training of the YOLOV4 algorithm are described. Also, the average loss charts during training and the metric evaluation of the trained model are available in the Appendix A-D section. The Appendix subsection consists of four categories and three subcategories, while the categories represent each training cycle of 6000, 8000, 10000, and 20000 max batch size, and the subcategories represent the Linear ReLU, and Mish activation functions. The final table of evaluation metrics can be seen in Tables 4 and 5.”

The following text was added in  Results subsection:

The results of the first and second training cycles i.e. are given in 4. Linear, ReLU, and Mish activation functions were used in 70/30,75/25, and 80/20 train/test ratios. All models were evaluated with mAP, F-1 score, Average IoU, Precision, Recall, and Average loss metrics. In the first part of the first training cycle for the Linear activation function with 70/30, 75/25, and 80/20 train test ratios, the following results were obtained: mAP was between 59.11% and 60.18%, F-1 score was in a range between 0.47 and 0.52, Average IoU varied 336between 28.1% and 29.98%, the Precision score was between 0.40 and 0.44 while the Recall and Average loss in between 0.58 and 0.64 and 2.4285 and 2.6178. Progress in obtaining the final amount of Average loss can be seen in the Loss charts given in Appendix A.1. The second part of the first training cycle for ReLU activation function evaluation metrics is changed for all tree train/test ratios. The mAP was between 94.16% and 95.12%, the F-1 score was between 0.81 and 0.87, while the Precision and Recall had similar values where the minimum value was 0.75 and the maximum was 0.93, the best Average IoU is 62.18% and the worst 58.21%. The Average loss did not change a lot and it was between 1.18 and 1.26. Charts of Average loss are presented in Appendix A.2. In the third part of the first cycle for the Mish activation function, the following results peak values are the following: mAP value of 94.25%, F-1 score of 0.86, Average IoU 61.52%, Precision 0.81, Recall 0.93 and Average loss in the amount of 1.2489. As in the previous two parts, average loss charts can be seen in Appendix A.3. For the fourth and fifth, parts of the second training cycle, the results of the obtained models are slightly better than in the first three parts. The maximum mAP value is 99.77% in favour of the ReLU activation function, while the minimum value is 91.27% in favour of the Linear activation function. The maximum value of the F-1 Score was on several occasions 0.97 for the ReLU and Mish activation functions. Which was also the peak value, while the minimum value was 0.59 for the Linear activation function. Charts of the training process can be seen in Appendix C in subsections C.1 and C.2. The results of the third and fourth training cycles i.e. 10,000 and 20,000 max batch size are given in 5. The Linear activation function is eliminated because of poor performance from the previous investigation. As in the last case, each training cycle of this research was conducted using the previous three train/test ratios. In the last training cycle, it is evident that the lowest average loss results were obtained compared to other training cycles. The results of mAP vary between 88.33% and 99.96%, the highest F-1 score was 1.00 while the lowest was 0.86. Both Precision and Recall ranged between 0.79 and 1. The highest score achieved for the Average IoU was 91.13%. The whole training process and average loss curve for the last two training cycles can be seen in Appendix D. Training process for Relu activation function is shown in Appendix D.1 and for Mish in Appendix D.2.”

In the discussion section:

“In the last training cycle, the ReLu and Mish activation functions were re-evaluated for all three train test ratios. All of the metrics are higher, which contributes to the very detection of returnable packaging. A possible explanation for this is the Mish activation function”

We have used blue highlight to note the changes made to the manuscript due to your comments. We hope you will be satisfied with our answers to the questions posed.

Kindest regards,
The Authors

Reviewer 4 Report

This paper focused on specifying the implementation details of YOLO to detect returnable packaging. Here are some major comments for improving this paper.

1. The whole writing style is too oral, resulting in the paper becoming more like a technical report. The English should be improved drastically. Abbreviations and formatting should be consistent throughout the paper, e.g., "You Only Look Once (YOLO)" on Line 1 and "You-Only-Look-Once (YOLO)" on Line 67.

2. Authors stated that "the correct choice of hyper-parameters and dataset size have significant influence on its detection and classification accuracy" (Line 10). The numerical results should be presented rather than using the qualitative vocabulary "significant influence".

3. From Line 82 to Line 85, it is really hard to follow and should be improved.

4. The motivation for this paper is unclear and needs to be rewritten. Also, the contribution or novelty of this paper should be emphasized to reflect the worth of your research.

5. There are too many useless references that destroy the integrity of your paper and reduce the space of the author's personal standpoint. Also, such many references give the reader the misconception that you have done nothing but cite. For example, was Figure 7 designed and drawn by the authors themselves, or was it produced by [56]? The references should be totally optimized.

Author Response

Dear reviewer,

The authors want to thank the reviewer for the time, effort, and constructive suggestions that have greatly improved the quality of this manuscript. The authors of this manuscript hope that the changes made in this manuscript will provide a suitable scientific contribution.

This paper focused on specifying the implementation details of YOLO to detect returnable packaging. Here are some major comments for improving this paper.

  1. The whole writing style is too oral, resulting in the paper becoming more like a technical report. The English should be improved drastically. Abbreviations and formatting should be consistent throughout the paper, e.g., "You Only Look Once (YOLO)" on Line 1 and "You-Only-Look-Once (YOLO)" on Line 67.

The authors have made an effort and have rewritten the manuscript. The authors do hope that they have improved the manuscript quality and that will be considered for publishing in this form. The English language has been improved and abbreviations were corrected to be consistent throughout the paper.

  1. Authors stated that "the correct choice of hyper-parameters and dataset size have significant influence on its detection and classification accuracy" (Line 10). The numerical results should be presented rather than using the qualitative vocabulary "significant influence".

The authors have rewritten the last two sentences of the abstract and in the revised version the best results in terms of highest mean Average precision and lowest average error are shown for specific combinations of hyperparameters (activation function and max batch sizes). Citing from the revised version of the manuscript (lines 7-11): “The conducted investigation showed that variation of hyperparameters (activation function and max batch sizes) have a significant influence on detection and classification accuracy with best results obtained in case of YOLO version 4 (YOLOV4) with Mish activation function and max batch size of 20000 that achieved the highest mAP of 99.96\% and lowest average error of 0.3643.

  1. From Line 82 to Line 85, it is really hard to follow and should be improved.

Line 83-101 was moved and modified TO 78-99 in revised version of the manuscript. Citing from the revised version of the manuscript:Based on related works and topics related to the detection of waste material, the YOLOV4 detection algorithm was additionally analyzed. There are several reasons for choosing the YOLO algorithm over other related algorithms for example, Kim et al. [22] applied YOLOV4 algorithm and achieved the most optimal ratio of mAP and frames per second (FPS) in real-time vehicle detection, while Faster-RCNN had the worst performance, i.e. it was the slowest and least accurate compared to other algorithms. Single Shot Detector (SSD) had highest FPS but significantly lower Precision than Faster-RCNN and YOLOV4. In terms of object localization, according to Ye et al. [21], it is evident that the YOLO detection algorithm compared to RCNN, RRPN, RetinaNet-H, RetinaNet-R, R-3Det, RCI & RC2, RPN, RRD and, ROI-Transformer, was the fastest with 60 FPS value compared to others. Robotic systems also have applications and impacts on reducing the amount of garbage in the environment. AI-controlled Outdoor Autonomous Trash-Collecting Robot was used for underwater garbage collection [23]. Several objects detection algorithms such as Mask-RCNN YOLOV4 and YOLOV4-tiny were compared and tested for underwater waste detection and classification. Results showed that the YOLO algorithm has had the highest mAP and detection speed compared to other used algorithms. The final but most important and main reason, besides the robustness and high rate of Precision and detection speed, is the size of the weights generated by the YOLO detection algorithm. Tian et al. [24] trained and several AI algorithms in which 4SP-YOLOV4 compared to Faster R-CNN, SSD, etc. has have a much smaller weight file size which can lead up to a possibility of a boarder range of the model implementation to printed circuit board-based microcontrollers such as Raspberry Pi, CaffeLatte, etc.

  1. The motivation for this paper is unclear and needs to be rewritten. Also, the contribution or novelty of this paper should be emphasized to reflect the worth of your research.

In the revised version of the manuscript, the authors have emphasized as much as possible the general idea/goal of this paper. Citing from the revised version of the manuscript (lines 100 – 114):The goal of this paper is to examine the YOLOV4 algorithm and achieve the optimal performance by varying activation function and max batch size hyperparameters on a custom-made, three-class (plastic, glass, and aluminum) dataset that contains 3788 images (2838 original images and 950 downloaded from Kaggle [25]). Based on the idea of this investigation and the extensive literature overview the following hypotheses questions arise:….

  1. Too many useless references destroy the integrity of your paper and reduce the space of the author's personal standpoint. Also, such many references give the reader the misconception that you have done nothing but cite. For example, was Figure 7 designed and drawn by the authors themselves, or was it produced by [56]? The references should be totally optimized.

The authors agree with the reviewers’ comments and some of the references were omitted in the revised version of the manuscript. The purpose of each reference that remains in the revised version of the manuscript is justified below. Figure 7 was not taken from other literature it was made by the authors of this manuscript from scratch.

Each reference used in this paper is explained below in the following form:

[Reference number] -Explanation

[1] Definition and description of waste management.

[2],  [3], [4], [5] – Examples of computer vision tasks such as image annotation, vehicle counting, activity recognition, and face detection.

[6], [7], [8], [9] – Examples of practical applications of object detection.

[10] – [21] – Examples of implementation of different deep learning, and machine learning (YOLO, CNN, ResNet 50, GA+DenseNet121, etc.) algorithms for detection/classification waste management. Overview of related investigations using artificial intelligence for waste management.

[22] – [24] Comparison of YOLO algorithm performance to other deep learning neural networks to show the advantages and disadvantages of the aforementioned algorithm. The comparison was demanded by one of the reviewers in the major revision process to justify why we have chosen the YOLOV4 algorithm in our investigation.

[25] – The reference for the dataset part which was used to create the dataset for this investigation.

[26] – [28] – Description of the dataset collection procedure. The related research shows examples of data collection and its importance.

[29] – [32] – The importance of dataset diversity to increase the performance of machine learning algorithms.

[33] – The influence of dataset size on the performance of artificial intelligence model.

[34]-[36] – Examples of augmentation techniques used in similar research.

[37] – OpenCV library

[38] – The review of imbalanced dataset classification and solutions to overcome this problem.

[39] – The importance of dataset quality and investigation of its influence on the performance of machine learning algorithms.

[40] – Reference to annotation tool used in this investigation. The name of the annotation tool: Visual Object Tagging Tool (VoTT).

[41] – [42] – The reference to dataset preparation techniques used in this research, as well as the example of its application (YOLO-based traffic counting system)

[43] – The reference explaining the output of VoTT software. Citing from the revised version of the manuscript: “…the first row of the column indicates the image class defined in the VoTT software, and second and third denote the center of the object along the x and y axes, fourth and fifth denote the width and height of the element [43].”

[44] – The reference used to example the inner workings of the YOLOV4 algorithm

[45] – The reference explaining the reliability coefficient determined while training the YOLOV4 algorithm

[46] –[49] The references show examples of YOLOV4 advantages. Reference [46] is an example that shows how multiple objects can be recognized in one frame. Reference [47, 48, 49] is an example of the detection principle and comparison between YOLOV4, R-CNN, and Fast R-CNN.

[50] – The reference shows the benefit of using the YOLOV4 algorithm.

[51], [52] – The references used to explain the Non-Max  Suppression procedure.

[53], [54] – The references used to describe in detail the Non-Max Suppression procedure.

 [55] – The reference explaining the improvements made in YOLO Version 4 algorithm

[56]-[60] – The references used to explain the intersection over union (IoU) metric.

[61] – The reference used to explain the mean average precision (mAP) metric.

[62] – The reference used to explain the precision and recall metric.

[63] – The reference used to explain the F1-Score metric.

[64] – The reference used to explain how YOLOV4 can struggle in a small object that appears in groups such as a flock of birds or a group of ants.

[65],[66] – The references used to explain the most commonly used train/test ratio in object detection algorithms.

[67] – The reference in which the mathematical equation for average training losses is defined.

[68] – The reference used to explain the flexibility of the Mish activation function.
